# Phage anti-CBASS and anti-Pycsar nucleases subvert bacterial immunity

Samuel J. Hobbs[1,2], Tanita Wein[3], Allen Lu[1,2], Benjamin R. Morehouse[1,2], Julia Schnabel[1,2], Azita Leavitt[3], Erez Yirmiya[3], Rotem Sorek[3] & Philip J. Kranzusch[1,2,4 ✉]

The cyclic oligonucleotide-based antiphage signalling system (CBASS) and the pyrimidine cyclase system for antiphage resistance (Pycsar) are antiphage defence systems in diverse bacteria that use cyclic nucleotide signals to induce cell death and prevent viral propagation[1,2]. Phages use several strategies to defeat host CRISPR and restriction-modification systems[3–10], but no mechanisms are known to evade CBASS and Pycsar immunity. Here we show that phages encode anti-CBASS (Acb) and anti-Pycsar (Apyc) proteins that counteract defence by specifically degrading cyclic nucleotide signals that activate host immunity. Using a biochemical screen of 57 phages in *Escherichia coli* and *Bacillus subtilis*, we discover Acb1 from phage T4 and Apyc1 from phage SBSphiJ as founding members of distinct families of immune evasion proteins. Crystal structures of Acb1 in complex with 3′3′-cyclic GMP–AMP define a mechanism of metal-independent hydrolysis 3′ of adenosine bases, enabling broad recognition and degradation of cyclic dinucleotide and trinucleotide CBASS signals. Structures of Apyc1 reveal a metal-dependent cyclic NMP phosphodiesterase that uses relaxed specificity to target Pycsar cyclic pyrimidine mononucleotide signals. We show that Acb1 and Apyc1 block downstream effector activation and protect from CBASS and Pycsar defence in vivo. Active Acb1 and Apyc1 enzymes are conserved in phylogenetically diverse phages, demonstrating that cleavage of host cyclic nucleotide signals is a key strategy of immune evasion in phage biology.

To determine how phages evade cyclic nucleotide-based bacterial immune systems, we developed a biochemical screen to analyse the stability of 11 distinct cyclic nucleotide signals during infection with 57 diverse phages (Fig. 1a and Supplementary Table 1). CBASS and Pycsar systems are widely distributed throughout the bacterial kingdom and are present in both Gram-negative and Gram-positive bacteria including *E. coli* and *B. subtilis*[1,2,11]. Lysates from uninfected laboratory strains of *E. coli* or *B. subtilis* readily hydrolyse common cyclic nucleotide signals including cyclic-di-GMP (cGG) and cyclic-di-AMP (cAA; Fig. 1b, c and Extended Data Figs. 1 and 2), consistent with known bacterial enzymes that regulate these signals during basal cellular function[12,13]. By contrast, CBASS and Pycsar antiphage signals including 3′3′-cyclic GMP–AMP (3′3′-cGAMP) and 3′,5′-cyclic CMP (cCMP) are exceptionally stable and remain intact following 20-h incubation in uninfected lysates. Strikingly, infection with diverse phages causes rapid hydrolysis of cyclic nucleotide signals specifically involved in immune defence (Fig. 1b, c). Lysates from cells infected with phage T4 and other closely related T-even coliphages degrade distinct classes of CBASS signals including cyclic dinucleotides 3′3′-cGAMP and 3′3′-cyclic UMP–AMP (cUA), and cyclic trinucleotides 3′3′3′-cyclic AMP–AMP–AMP (cAAA) and 3′3′3′-cyclic AMP-AMP-GMP (cAAG). Likewise, lysates from cells infected with the SBSphiJ family of *B. subtilis* phages rapidly degrade the Pycsar signals cCMP and cUMP (Fig. 1b, c). Except for the rare CBASS dinucleotides 3′3′-c-di-UMP and 3′2′-cGAMP, all known

cyclic nucleotide signals used in CBASS or Pycsar immune defence were susceptible to degradation by at least one phage (Fig. 1c).

## Phages encode immune evasion nucleases

Rapid degradation of cyclic nucleotide signals used in host immunity suggests that phages encode proteins dedicated to CBASS and Pycsar evasion. To define anti-CBASS (Acb) and anti-Pycsar (Apyc) proteins, we first focused on phage T4 and used an activity-guided fractionation and mass spectrometry approach to identify candidate Acb proteins responsible for 3′3′-cGAMP cleavage (Fig. 2a and Extended Data Fig. 3a). In vitro screening of each candidate demonstrated that the uncharacterized T4 gene *57B* encodes a protein that degrades the CBASS signal 3′3′-cGAMP (Extended Data Fig. 3b, c), and we named this anti-CBASS protein Acb1 (GenBank accession number NP_049750.1). Recombinant T4 Acb1 rapidly degrades the CBASS signals 3′3′-cGAMP, cUA and cAAA, but does not cleave cGG, demonstrating that Acb1 is responsible for the broad cyclic nucleotide hydrolysis activity observed in T4-infected cell lysate (Fig. 2b, c and Extended Data Fig. 4). We next identified candidate Apyc proteins within SBSphiJ-family phages that cleaved cCMP in our biochemical screen. Genome sequencing and comparative bioinformatic analysis of eight closely related SBSphiJ-family phages revealed two genomic regions present exclusively in phages

[1]Department of Microbiology, Harvard Medical School, Boston, MA, USA. [2]Department of Cancer Immunology and Virology, Dana-Farber Cancer Institute, Boston, MA, USA. [3]Department of Molecular Genetics, Weizmann Institute of Science, Rehovot, Israel. [4]Parker Institute for Cancer Immunotherapy at Dana-Farber Cancer Institute, Boston, MA, USA. ✉e-mail: philip_kranzusch@dfci.harvard.edu

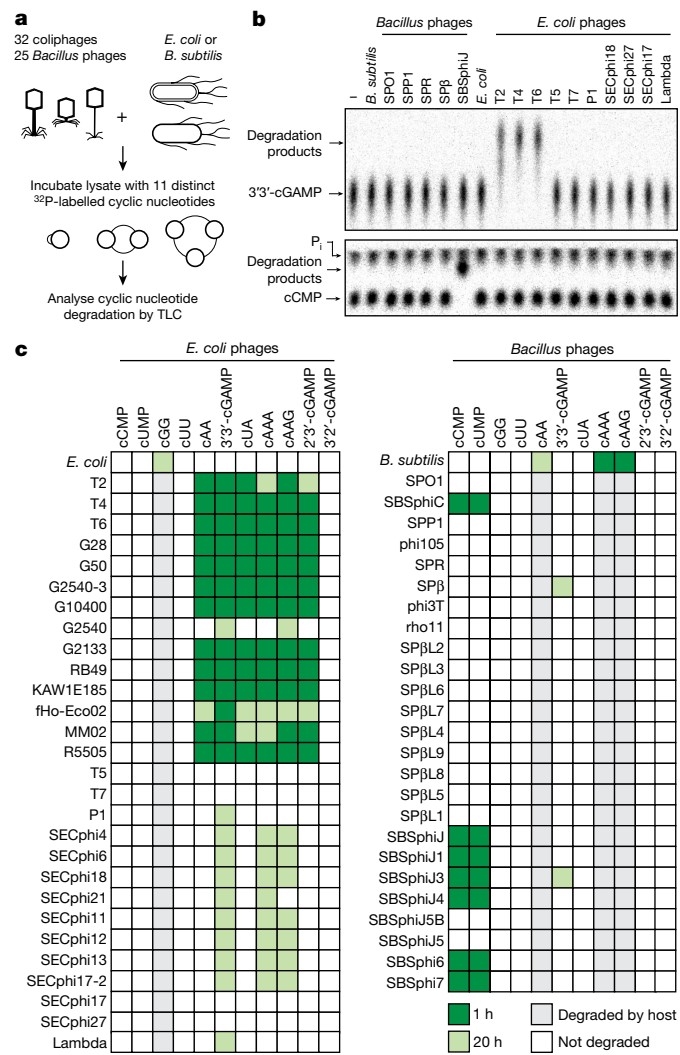

**Fig. 1 | Phages selectively degrade cyclic nucleotide signals used in host defence. a**, Schematic depicting a screen of cyclic nucleotide degradation activity in phage-infected lysates using thin-layer chromatography (TLC). **b**, Representative TLC assays depicting cleavage of 3′3′-cGAMP following infection by T2, T4 and T6 phages, or cleavage of cCMP following infection with SBSphiJ phage. Data are representative of at least two independent replicates. Pᵢ, inorganic phosphate; −, buffer only control. **c**, Summary of the complete results of the screen in **b**, with four phages closely related to T5 omitted for clarity (see Supplementary Table 1 for complete list of phages). The green shading represents the incubation times indicated in the key. T4-related phages degrade diverse CBASS signals and SBSphiJ-related phages degrade diverse Pycsar signals.

capable of degrading cCMP (Fig. 2d and Extended Data Fig. 5a). We used structure prediction to analyse each protein encoded in these regions and identified that the uncharacterized SBSphiJ gene *147* encodes a protein with predicted homology to known metallo β-lactamase (MBL) fold RNase and phosphodiesterase enzymes (Fig. 2d and Extended Data Fig. 5a, b). Recombinant protein produced from gene *147* rapidly degrades the Pycsar signals cCMP and cUMP (Fig. 2e, f and Extended Data Fig. 5c–f), and we named this anti-Pycsar protein Apyc1 (European Nucleotide Archive genome accession number ERS1981056). SBSphiJ Apyc1 efficiently hydrolyses a wide range of cyclic mononucleotides (Fig. 2f), exhibiting an atypically relaxed nucleobase specificity that enables targeting of cyclic pyrimidine signals used in Pycsar immunity.

Immune evasion genes frequently cluster together in the genomes of phages to form anti-defence islands[7,14]. Consistent with a role in CBASS evasion, T4 Acb1 is encoded adjacent to internal protein I (ipI), a phage inhibitor required to evade the *E. coli* restriction enzyme gmrS/gmrD

that recognizes glucosylated cytosine bases present in T4 genomic DNA (ref. [15]; Fig. 2g). *Apyc1* is the first identified anti-defence gene in SBSphiJ, limiting comparative analysis with other genes in this phage. However, Apyc1 is encoded adjacent to a series of small proteins of unknown function, suggesting that this variable locus in SBSphiJ-family phages may contribute to evasion of other antiphage defence systems (Fig. 2h). To discover further Acb and Apyc proteins, we searched for proteins related to Acb1 and Apyc1 within phage genomes and prophage sequences (Fig. 2i, j). Analysis of T4 Acb1 identified 281 related protein sequences with about 97% predicted to be of phage origin. We cloned and tested a further 9 *acb1* genes and observed that each recombinant Acb1 protein efficiently cleaved the CBASS signals 3′3′-cGAMP and cAAA (Fig. 2i and Extended Data Fig. 6a). We identified 107 proteins related to Apyc1 present in phage genomes (Fig. 2j) and also found many closely related bacterial proteins encoded in diverse bacterial orders (Extended Data Fig. 6b). Similar to SBSphiJ Apyc1, closely related phage and bacterial Apyc1-like proteins cleaved cyclic mononucleotides with broad specificity (Fig. 2j and Extended Data Fig. 6c). By contrast, the closely related *B. subtilis* enzymes YhfI (GenBank accession number NP_388905.1) and MBL phosphodiesterase (GenBank accession number WP_013351727.1) exhibited a strong preference for cAMP/cGMP over cCMP/cUMP cleavage, confirming that relaxed nucleotide specificity and Pycsar signal degradation are unique to Apyc1 and not general features of MBL phosphodiesterase enzymes (Extended Data Fig. 6d). The observation of Apyc1 homologues encoded in bacteria may be explained by the presence of cryptic prophages present in bacterial genomes, but also raises the intriguing possibility that host Apyc1 enzymes may play a role in regulating Pycsar defence or other cNMP-based signalling systems. In total, our analysis identified 273 Acb1 and 107 Apyc1 phage proteins, demonstrating that cyclic nucleotide-degrading enzymes constitute a widespread form of anti-CBASS and anti-Pycsar evasion.

## Mechanisms of cyclic nucleotide cleavage

We next determined crystal structures of Acb1 to define the mechanism of anti-CBASS evasion. Structures of Acb1 from the *Erwinia* phage FBB1 in the apo state (1.1 Å) and in complex with 3′3′-cGAMP (1.2 Å) reveal that Acb1 adopts a compact 2H phosphoesterase fold with six central β-strands that form a U-shaped ligand-binding pocket (Fig. 3a, Extended Data Fig. 7a and Supplementary Table 2). On substrate recognition, the flexible carboxy-terminal residues 145–152 form an ordered lid that closes over the top of the captured 3′3′-cGAMP ligand (Fig. 3a and Extended Data Fig. 7b). Acb1 ligand recognition is primarily independent of base identity, with the conserved aromatic residues Y12, W74, F107 and W147 forming stacking interactions with the face of each nucleobase (Fig. 3b). However, base-specific contact occurs between E141 and the 3′3′-cGAMP adenosine N6 position, explaining why at least one adenosine is required for cleavage (Fig. 2c and Extended Data Fig. 7c). Although overall lack of sequence-specific contacts allows Acb1 to target a broad range of CBASS cyclic nucleotide signals, the Acb1 binding pocket can accommodate only cyclic dinucleotide or trinucleotide species. Structural clashes prevent recognition of larger cyclic oligonucleotides with >3 bases, and we confirmed that Acb1 is unable to degrade cyclic tetra-adenylate (cA₄) rings common in type III clustered regularly interspaced short palindromic repeats (CRISPR) immunity[16,17] (Extended Data Fig. 7d). Acb1–nucleotide interactions contort 3′3′-cGAMP into a highly strained conformation in which the adenosine base is rotated about 65° relative to the in-solution or receptor-bound conformation, repositioning the 2′ OH for attack on the 3′–5′ bond[18,19] (Fig. 3c). In the Acb1–3′3′-cGAMP structure, the scissile phosphate is positioned over an active-site HxT/HxT tetrad (H44, T46, H113, T115) for acid–base catalysis and the ligand is fully hydrolysed into the linear product G[3′–5′]pAp[3′] (GpAp) (Fig. 3d and Extended Data Fig. 7e). We tracked cleavage reactions in vitro using high-performance liquid chromatography (HPLC) and confirmed that Acb1 cleaves 3′ of adenosine residues in a two-step, metal-independent reaction that proceeds through a cyclic

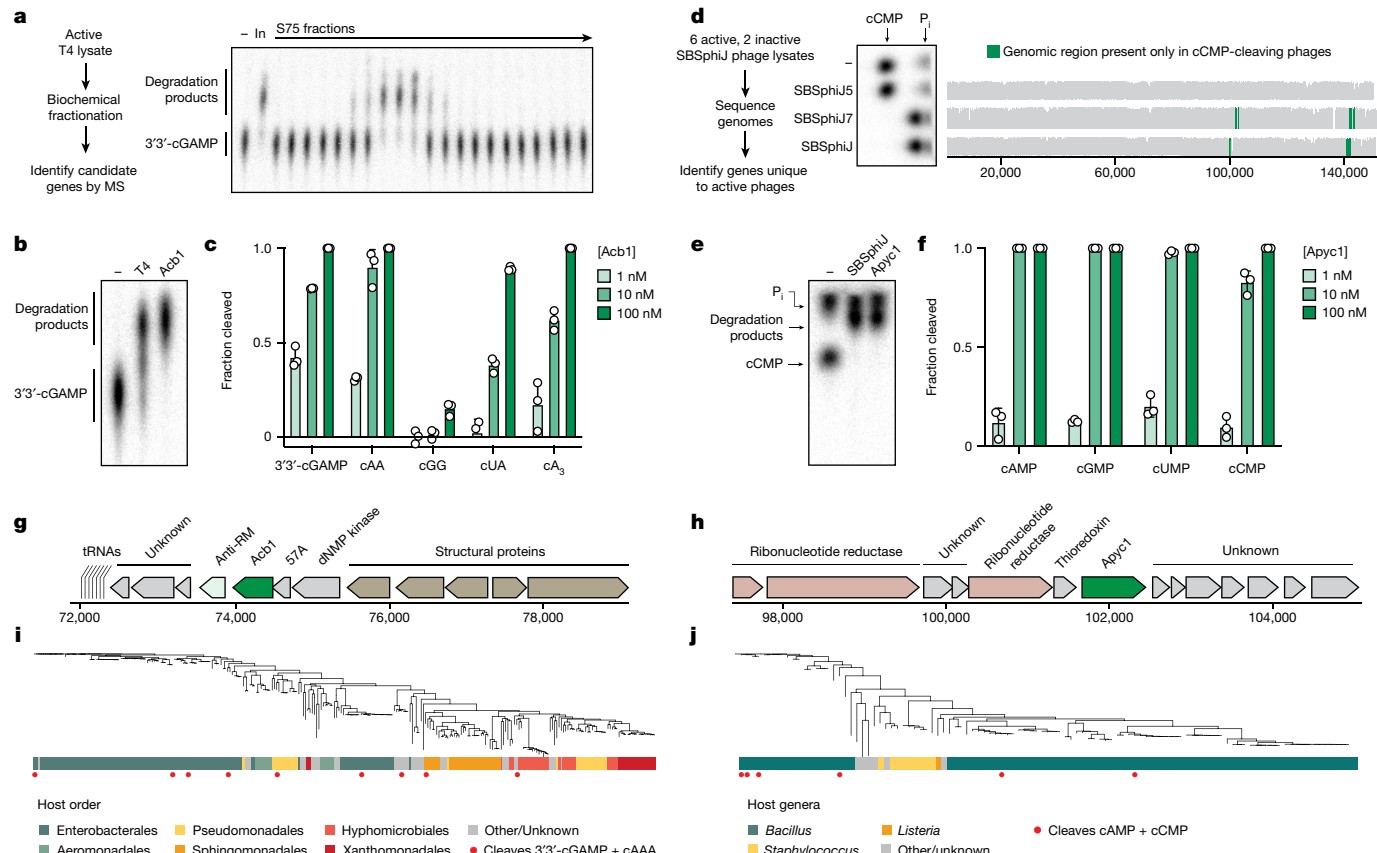

**Fig. 2 | Distinct viral nucleases target CBASS and Pycsar immune signals.**
**a**, Schematic and representative example of activity-guided biochemical fractionation and mass spectrometry (MS) to identify Acb1 candidate genes from phage T4. Fractions were collected from an S75 size-exclusion column and tested for 3′3′-cGAMP activity. In, crude lysate input. Data are representative of two independent experiments. **b**, Comparison of 3′3′-cGAMP cleavage by T4 lysate and recombinant Acb1. Data are representative of three independent experiments. **c**, Summary of HPLC analysis testing Acb1 substrate specificity (20-min incubation). Acb1 cleaves dinucleotide and trinucleotide CBASS signals containing one or more AMP. Data are presented as mean ± s.d. from $n = 3$ independent experiments. **d**, Bioinformatic analysis identifies candidate Apyc1 genes from genomic regions exclusive to cCMP-cleaving phages. TLC data are representative of two independent

experiments. **e**, Comparison of cCMP cleavage by SBSphiJ lysate and recombinant Apyc1. Data are representative of three independent experiments. **f**, Summary of HPLC analysis testing Apyc1 substrate specificity (20-min incubation). Apyc1 cleaves all cNMP signals with equal efficiency. Data are presented as mean ± s.d. from $n = 3$ independent experiments. **g**, **h**, Schematics showing genes neighbouring T4 Acb1 (**g**) and SBSphiJ Apyc1 (**h**); dNMP, deoxyribonucleoside monophosphate. **i**, Phylogenetic tree showing T4 Acb1 and 271 related protein sequences from phages, including 112 sequences derived from prophages. Colour strips indicate the order of the bacterial host. Red circles indicate proteins tested for cleavage of 3′3′-cGAMP and cAAA. **j**, Phylogenetic tree displaying SBSphiJ Apyc1 and 106 related protein sequences from phages. Colour strips indicate the genus of the bacterial host. Red circles indicate proteins tested for cleavage of cAMP and cCMP.

phosphate intermediate (Extended Data Fig. 7f). Substitutions of conserved active-site and nucleotide-coordinating residues disrupt enzyme function and highlight the critical role for contacts stabilizing the rotated adenine base in Acb1 cyclic nucleotide cleavage (Fig. 3e).

To compare mechanisms of anti-CBASS and anti-Pycsar evasion, we determined the crystal structure of Apyc1 from the phage Bsp38 (2.7 Å) as well as structures of *Paenibacillus* Apyc1 proteins (1.5 Å and 1.8 Å). These structures confirm that Apyc1 is a member of the class II phosphodiesterase enzymes, which exhibit an MBL fold and have no structural or mechanistic homology to Acb1 (ref. [20]; Extended Data Fig. 8a, b and Supplementary Table 2). Similar to other structurally characterized class II phosphodiesterases such as *B. subtilis* YhfI, yeast *Saccharomyces cerevisiae* PDE1 or widely distributed RNase Z proteins[21,22], Apyc1 is a homodimer with a highly conserved HxHxDH motif that coordinates two $Zn^{2+}$ ions that bind phosphate groups to position cyclic nucleotides for cleavage (Extended Data Fig. 8a–c). In a structure of *Paenibacillus* Apyc1 co-crystallized in the presence of nonhydrolysable cAMP, we observed strong electron density near the $Zn^{2+}$ ions and more diffuse density in the nucleobase pocket, consistent with specific coordination of the phosphate and ribose backbone of cyclic mononucleotides and weaker nucleobase specificity within the enzyme

active site (Extended Data Fig. 8d). Structural comparison of Apyc1 and *B. subtilis* YhfI also reveals that Apyc1 enzymes contain an extended loop that reaches into the nucleotide-binding pocket, potentially enabling stable binding of smaller cyclic pyrimidine substrates (Extended Data Fig. 8b). We confirmed the critical role for Apyc1 metal-coordinating residues and identified E74 and Y112 from the opposing protomer as further catalytic residues required for cCMP hydrolysis and release of the reaction product 5′-CMP (Extended Data Fig. 8e, f). Together, these findings demonstrate that Acb1 and Apyc1 constitute separate families of immune evasion proteins and explain the distinct reaction mechanisms that degrade CBASS or Pycsar cyclic nucleotide signals (Fig. 3f).

## Acb1 and Apyc1 subvert host immunity

CBASS and Pycsar antiphage defence requires cyclic nucleotide-dependent activation of downstream effector proteins that induce cell death[1,2,23–26]. Using a panel of CBASS nuclease and phospholipase effectors from *Vibrio cholerae*, *Enterobacter cloacae* and *Burkholderia pseudomallei*, we reconstituted CBASS signalling in vitro and observed that Acb1 potently inhibited activation of both cyclic dinucleotide- and cyclic trinucleotide-responsive effectors[23] (Fig. 4a and Extended Data Fig. 9a, b).

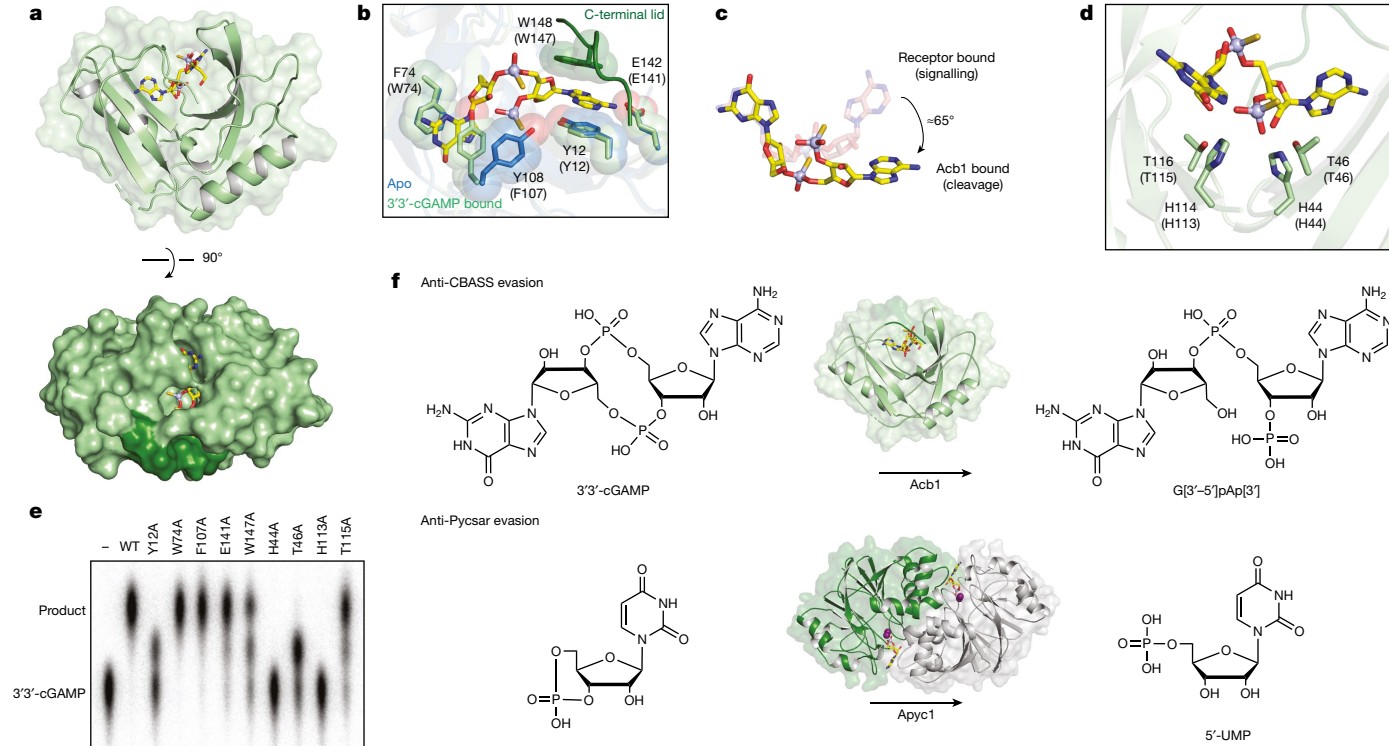

**Fig. 3 | Structural basis of Acb1 3'3'-cGAMP degradation. a**, Overview of Acb1 from *Erwinia* phage FBB1 in complex with a hydrolysis-resistant phosphorothioate analogue of 3'3'-cGAMP. In the surface representation, C-terminal lid residues are coloured dark green. **b**, Detailed view of residues interacting with the bases of 3'3'-cGAMP. Parentheses indicate equivalent position in T4 Acb1. **c**, Conformations of 3'3'-cGAMP bound to STING (Protein Data Bank (PDB): 5CFM) or Acb1. **d**, Detailed view of catalytic residues. Parentheses indicate equivalent position in T4 Acb1. **e**, Thin-layer chromatography analysis of 3'3'-cGAMP cleavage by T4 Acb1 point mutants. Data are representative of three independent experiments. **f**, Schematic of reactions catalysed by Acb1 and Apyc1.

Likewise, Apyc1 enzymatic activity abolished cUMP-dependent activation of the Pycsar NADase effector PycTIR (ref. [2]; Fig. 4b). The activities of Acb1 and Apyc1 are specific to CBASS or Pycsar signalling, demonstrating that anti-CBASS and anti-Pycsar immune evasion proteins are dedicated to each class of antiphage defence system (Fig. 4a, b and Extended Data Fig. 9a). Tracking Acb1 and Apyc1 activity during infection, we observed that cyclic nucleotide degradation activity begins about 15 min into phage T4 infection and about 30 min into phage SBSphiJ infection, coinciding with the known late onset of CBASS and Pycsar antiphage cell death responses[1,2] (Extended Data Fig. 9c, d). Acb1 expression in *E. coli* inhibited CBASS-mediated cell death in vivo, suggesting that immune evasion proteins can protect phages from premature abortive infection responses (Extended Data Fig. 9e).

To define the importance of degradation of cyclic nucleotide immune signals during phage infection, we infected *E. coli* expressing complete CBASS and Pycsar defence operons and quantified the effect of Acb1 and Apyc1 expression on phage replication. In the presence of an active type III CBASS operon from *E. coli* KTE188, Acb1 expression significantly boosted infectivity of the normally susceptible phage P1 by about 1.5 log (Fig. 4c). Likewise, expression of Apyc1 in *E. coli* disrupted Pycsar defence and completely rescued growth of phage T5, demonstrating that Acb1 and Apyc1 are sufficient to counteract host CBASS and Pycsar defence (Fig. 4d). To determine whether cyclic nucleotide degradation is necessary for immune evasion, we next focused on engineering a mutant phage lacking the ability to cleave immune nucleotide signals. Robust approaches do not yet exist for genetic manipulation of *B. subtilis* phages, and analysis of *apyc1*-deletion viruses will therefore be a focus of future research. However, we were able to use recent advances in coliphage engineering to create a phage T4 mutant virus lacking functional Acb1 (phage T4 Δ*acb1*) (Extended Data Fig. 10a). *E. coli* cells infected with phage T4 Δ*acb1* do not hydrolyse 3'3'-cGAMP, confirming

that Acb1 is essential for viral degradation of CBASS immune cyclic nucleotides (Extended Data Fig. 10b). In the absence of functional CBASS defence, phage T4 and phage T4 Δ*acb1* grow equally well, revealing that Acb1 is not required for normal replication in *E. coli* (Fig. 4e, f and Extended Data Fig. 10c, d). In contrast, growth of phage T4 Δ*acb1* is specifically impaired in the presence of active CBASS immunity with the mutant virus exhibiting a >300-fold defect in viral replication compared to wild-type phage T4 (Fig. 4e, f and Extended Data Fig. 10c, d). These results demonstrate that viral nucleases are critical for evasion of cyclic nucleotide-mediated phage defence.

Together, our data define Acb1 and Apyc1 as founding members of families of anti-CBASS and anti-Pycsar immune evasion proteins that allow phages to selectively hydrolyse cyclic nucleotide immune signals used for host defence. No single phage could degrade all cyclic nucleotide immune signals, revealing that diversification of cyclic nucleotide signals between CBASS and Pycsar systems is a key host adaptation to maintain successful antiphage defence[2,11]. Acb1 and Apyc1 join a growing collection of viral nuclease enzymes dedicated to immune evasion, including phage ring nucleases that degrade cA$_4$ and cA$_6$ signals used in type III CRISPR immunity[27,28] and poxin enzymes that degrade 2'3'-cGAMP to inhibit cyclic GMP–AMP synthase (cGAS)–stimulator of interferon genes (STING) signalling in animals[29]. Each of these viral enzymes is structurally distinct, demonstrating at least four separate instances of prokaryotic and eukaryotic viral evolution to degrade host cyclic nucleotide immune signals. The broad specificity of Acb1 allows evasion of diverse CBASS operons with a single gene, and the ability of Acb1 to cleave cyclic trinucleotide species suggests that this enzyme may also enable evasion of type III CRISPR systems that use cAAA signals. Notably, Acb1 is unable to cleave the non-canonical 2'–5' linkage in the CBASS signalling molecule 3'2'-cGAMP (ref. [30]), mirroring the recent demonstration that 3'2'-cGAMP signalling in animals enables resistance

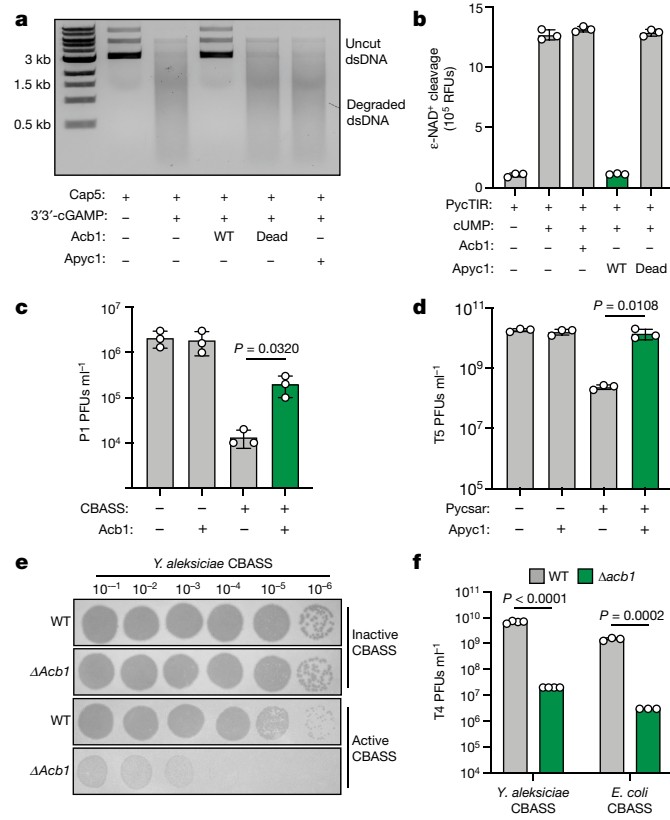

**Fig. 4 | Acb1 and Apyc1 disrupt CBASS and Pycsar host defence. a**, Agarose gel analysis of uncut plasmid DNA incubated with the CBASS effector Cap5 and 3'3'-cGAMP that was treated with wild-type (WT) Acb1, catalytically inactive Acb1-H44A/H113A or WT Apyc1. Data are representative of three independent experiments. For unprocessed gels, see Supplementary Fig. 1. ds, double-stranded DNA. **b**, Release of fluorescent substrate from an NAD[+] analogue incubated with the Pycsar effector PycTIR and cUMP that was treated with WT Acb1, WT Apyc1 or catalytically inactive Apyc1-H64A/H66A/H69A. Data are presented as mean ± s.d. from $n = 3$ independent experiments. RFUs, relative fluorescence units. **c**, *E. coli* carrying plasmids encoding a type III CBASS operon from *E. coli* KTE188 and/or T4 Acb1 were challenged with serial dilutions of P1 phage. Data are presented as mean ± s.d. from $n = 3$ independent experiments. PFUs, plaque-forming units. **d**, *E. coli* carrying plasmids encoding a Pycsar operon and/or SBSphiJ Apyc1 were challenged with serial dilutions of T5 phage. Data are presented as mean ± s.d. from $n = 3$ independent experiments. **e**, Representative plaque assays of *E. coli* carrying a plasmid encoding an active or catalytically inactive CBASS operon from *Yersinia aleksiciae* and challenged with WT phage T4 or phage T4 engineered to remove Acb1 (Δ*acb1*). **f**, Summary of plaque assay results of WT or Δ*acb1* phage T4 infection of *E. coli* carrying CBASS operons from *Y. aleksiciae* or *E. coli*. Data are presented as mean ± s.d. from $n = 4$ (*Y. aleksiciae* operon) or $n = 3$ (*E. coli* operon) technical replicates and are representative of at least 3 biologically independent experiments. Statistical significance in **c**, **d** and **f** was determined using an unpaired two-tailed *t*-test.

to poxin enzymes[31]. The large diversity of >180 possible nucleotide signals proposed to exist in antiphage defence suggests that in addition to signal degradation, phages may encode Acb and Apyc proteins that target alternative components of CBASS or Pycsar immunity. Overall, our results define viral nucleases as a widespread mechanism of CBASS and Pycsar immune evasion and reveal the role of viral proteins in driving evolution of cyclic nucleotide-based immune defence systems.

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

## Methods

### Bacterial strains and phages
*E. coli* strain MG1655 (ATCC 47076) and *B. subtilis* BEST7003 (ref. [32]) were grown in magnesium–manganese broth (MMB; lysogeny broth supplemented with 0.1 mM $MnCl_2$ and 5 mM $MgCl_2$) with or without 0.5% agar at 37 °C or 30 °C, respectively. Whenever applicable, media were supplemented with ampicillin (100 μg $ml^{-1}$), chloramphenicol (34 μg $ml^{-1}$) or kanamycin (50 μg $ml^{-1}$) to ensure the maintenance of plasmids. Phage isolation was performed as previously described[33]. In general, phage infections were performed in MMB media at 37 °C for *E. coli* MG1655 and at 30 °C for *B. subtilis* and phages were propagated by picking a single phage plaque into a liquid culture grown to an optical density at 600 nm ($OD_{600}$) of 0.3 in MMB medium until culture collapse. The culture was then centrifuged for 10 min at 3,200*g*, and the supernatant was filtered through a 0.2-μm filter. The titre of the lysate was determined using the small-drop plaque assay method as described previously[34].

### Recombinant protein expression and purification
Acb1, Apyc1, cGAS/DncV-like nucleotidyltransferases (CD-NTase), cGAS-like receptors and effector proteins were purified from *E. coli* as previously described[11,23,31,35]. Briefly, genes were cloned from synthetic DNA fragments (Integrated DNA Technologies) into custom pET expression vectors containing amino-terminal 6×His-SUMO2 or 6×His-MBP-SUMO2 tags by Gibson assembly using HiFi DNA Assembly Master Mix (NEB)[35]. Expression plasmids were transformed into BL21(DE3) RIL cells (Agilent) and plated onto MDG media (1.5% Bacto agar, 0.5% glucose, 25 mM $Na_2HPO_4$, 25 mM $KH_2PO_4$, 50 mM $NH_4Cl$, 5 mM $Na_2SO_4$, 0.25% aspartic acid, 2–50 μM trace metals, 100 μg $ml^{-1}$ ampicillin, 34 μg $ml^{-1}$ chloramphenicol). After overnight incubation at 37 °C, three colonies were used to inoculate a 30-ml MDG starter culture for 16 h (37 °C, 230 r.p.m.). M9ZB expression cultures of 1 l in volume (47.8 mM $Na_2HPO_4$, 22 mM $KH_2PO_4$, 18.7 mM $NH_4Cl$, 85.6 mM NaCl, 1% casamino acids, 0.5% glycerol, 2 mM $MgSO_4$, 2–50 μM trace metals, 100 μg $ml^{-1}$ ampicillin, 34 μg $ml^{-1}$ chloramphenicol) were then inoculated with 15 ml MDG starter culture and grown (37 °C, 230 r.p.m.) to an $OD_{600}$ of 2.5 before induction with 0.5 mM isopropyl-β-ᴅ-thiogalactoside (IPTG) for 16 h (16 °C, 230 r.p.m.). For WT Apyc1 protein, bacteria were grown in 2YT media (16 g $l^{-1}$ Bacto tryptone, 10 g $l^{-1}$ yeast extract, 5 g $l^{-1}$ NaCl, pH 7.0) for both starter and expression cultures and grown to an $OD_{600}$ of 1.5 before induction with 0.5 mM IPTG for 16 h (16 °C, 230 r.p.m.). Selenomethionine-labelled protein was prepared as previously described[29] by expressing 1 l cultures in modified M9ZB media (47.8 mM $Na_2HPO_4$, 22 mM $KH_2PO_4$, 18.7 mM $NH_4Cl$, 85.6 mM NaCl, 0.4% glucose, 2 mM $MgSO_4$, 2–50 μM trace metals, 1 μg $ml^{-1}$ thiamine, 100 μg $ml^{-1}$ ampicillin, 34 μg $ml^{-1}$ chloramphenicol) and allowing the cultures to grow to an $OD_{600}$ of 0.8 before supplementation with ʟ-amino acids (50 mg $ml^{-1}$ leucine, isoleucine, valine; 100 mg $ml^{-1}$ of phenylalanine, lysine, threonine; 75 mg $ml^{-1}$ selenomethionine) and induction with 0.5 mM IPTG for 16 h (16 °C, 230 r.p.m.).

After overnight expression, cell pellets were collected by centrifugation and then resuspended and lysed by sonication in 50 ml lysis buffer (20 mM HEPES-KOH pH 7.5, 400 mM NaCl, 10% glycerol, 30 mM imidazole, 1 mM TCEP). Lysate was clarified by centrifugation at 50,000*g* for 30 min, supernatant was poured over 8 ml Ni-NTA resin (Qiagen), resin was washed with 35 ml lysis buffer supplemented with 1 M NaCl, and protein was eluted with 10 ml lysis buffer supplemented with 300 mM imidazole. Samples were then dialysed overnight in dialysis tubing with a 14 kDa molecular weight cutoff (Ward's Science), and SUMO2 tag cleavage was carried out with recombinant human SENP2 protease as previously described[35]. Proteins used for crystallography were dialysed overnight at 4 °C in dialysis buffer (20 mM HEPES-KOH pH 7.5, 250 mM KCl, 1 mM TCEP), and then purified further by size-exclusion chromatography using a 16/600 Superdex 75 column (Cytiva), whereas proteins used for biochemical assays were dialysed in dialysis buffer supplemented with 10% glycerol. Purified proteins were concentrated to >15 mg $ml^{-1}$ using 10-kDa MWCO centrifugal filter units (Millipore Sigma), aliquoted, flash frozen in liquid nitrogen and stored at −80 °C.

### Thin-layer chromatography
Thin-layer chromatography was used to analyse cyclic nucleotide degradation as previously described[29]. Cyclic nucleotides were synthesized using the following purified recombinant enzymes: *V. cholerae* DncV (ref. [11]): cAA, 3′3′-cGAMP, cGG; *E. cloacae* CdnD (ref. [23]): cAAA, cAAG; *Rhodothermus marinus* CdnE (ref. [11]): cUA; *Y. aleksiciae* CdnE (ref. [36]): 3′3′-cUU; *Drosophila eugracilis* cGLR1 (ref. [31]): 3′2′-cGAMP; *Mus musculus* cGAS (ref. [35]): 2′3′-cGAMP; *E. coli* PycC (ref. [2]): cCMP; *Burkholderia cepacia* PycC (ref. [2]): cUMP. Synthesis reactions were performed at 37 °C for 20 h, and consisted of 2.5 μM appropriate enzyme, 25 μM appropriate nucleoside triphosphates (NTPs), trace amounts of α-$^{32}$P-labelled NTP, 100 mM KCl, 1 mM dithiothreitol (DTT), 5 mM $MgCl_2$, 1 mM $MnCl_2$ and 50 mM Tris-HCl pH 7.5 (DncV, cGLR1, cGAS) or pH 9.0 (all other enzymes) in a final volume of 40 μl. Unincorporated NTPs were digested by addition of 1 μl Quick CIP (NEB) followed by incubation at 37 °C for 30 min and heat inactivation at 95 °C for 2 min. Synthesis reactions were then used as inputs for downstream degradation reactions, which were carried out at 37 °C in 10-μl mixtures composed of 1 μl of a 10× recombinant enzyme stock or cellular lysate, 0.25–0.5 μl of the appropriate synthesis reaction (about 1–2 μM α-$^{32}$P-labelled cyclic nucleotide), 50 mM Tris-HCl pH 7.5, 10 mM KCl and 1 mM TCEP. After 5–20-min incubation (unless indicated otherwise), 0.5 μl volumes of reactions were spotted on a 20 cm × 20 cm PEI cellulose thin-layer chromatography plate (Sigma Aldrich) and developed in 1.5 M $KH_2PO_4$ (pH 3.8) buffer for 45 min. Plates were dried at room temperature, exposed to a storage phosphor screen, and detected with a Typhoon Trio Variable Mode Imager System (GE Healthcare).

### Cell lysate preparation
Overnight cultures of *E. coli* or *B. subtilis* were diluted 1:100 in 250 ml MMB medium and grown at 37 °C for *E. coli* and 30 °C for *B. subtilis* (250 r.p.m.) until reaching an $OD_{600}$ of 0.3. The cultures were infected with phages (Supplementary Table 1) at a final multiplicity of infection of 2. Samples of infected cells were taken before culture collapse (for time points, see Supplementary Table 1). Samples of 5 ml in volume were taken and centrifuged for 5 min at 3,200*g* and 4 °C. The culture pellets were flash frozen using dry ice and ethanol. *E. coli* pellets were resuspended in 250 μl of a lysis buffer containing 20 mM HEPES-KOH pH 7.5, 150 mM NaCl, 5 mM $MgCl_2$, 1 mM $MnCl_2$, 1 mM DTT, 10% glycerol and 1% NP-40, and incubated at room temperature for 30 min with occasional vortexing. *Bacillus* pellets were first treated with T4 lysozyme (ThermoFisher) at 1 mg $ml^{-1}$ in PBS at 37 °C for 10 min, followed by addition of 400 μl of *E. coli* lysis buffer and 30-min incubation at room temperature. Samples were clarified by centrifugation for 5 min at 17,000*g* at 4 °C, and the supernatant was aliquoted and flash frozen in liquid nitrogen, and stored at −80 °C.

### T4 Acb1 activity-guided fractionation, mass spectrometry analysis and candidate screen
Overnight *E. coli* MG1655 cultures were diluted 1:100 into a volume of 2 l MMB and grown for about 1 h to an $OD_{600}$ of 0.3–0.5. Phage T4 was added at a multiplicity of infection of 2, and cells were collected 25 min post infection by centrifugation for 20 min at 3,200*g*. Infected cell pellets were resuspended in 40 ml of lysis buffer consisting of 20 mM HEPES-KOH pH 7.5, 150 mM KCl, 1 mM DTT, 5 mM $MgCl_2$, 1 mM $MnCl_2$, 10% glycerol and 1% NP-40 and incubated at room temperature for 30 min with occasional vortexing. Lysates were clarified by centrifugation at 20,000*g* for 15 min at 4 °C and fractionated by ion-exchange chromatography using a 5-ml HiTrap SP column and a gradient of 0.05–1.0 M NaCl. Active ion-exchange fractions were pooled, concentrated,

and further separated with a 10/300 Superdex 75 column (Cytiva). In a separate approach, $(NH_4)_2SO_4$ was added to clarified lysates to a final concentration of 30%, and precipitated proteins were removed by centrifugation at 20,000$g$ for 15 min. The soluble fraction was then separated using hydrophobic interaction chromatography using a 5-ml phenyl column (Cytiva) and a gradient of 1–0.0 M $(NH_4)_2SO_4$. Active fractions were pooled, concentrated, and further separated with a 10/300 Superdex 200 column (Cytiva). For each enrichment scheme, phage T4 proteins enriched in fractions with the highest activity relative to neighbouring inactive fractions were quantified by label-free mass spectrometry as previously described[29].

Phage T4 genes identified by biochemical fractionation and mass spectrometry were amplified from genomic T4 DNA isolated from infected *E. coli* using a Qiagen DNeasy Blood and Tissue kit as described previously[37]. Candidate genes were PCR amplified using Q5 DNA polymerase (NEB) and primers designed to incorporate a 49-base-pair sequence containing a T7 promoter and a ribosome-binding site upstream of the amplified candidate gene according to the NEB cell-free *E. coli* protein synthesis system instructions (NEB). PCR products were purified using a PCR clean-up kit (Qiagen) and translated using the *E. coli* protein synthesis system kit (NEB). A 1 μl volume of each translation reaction was used to test for 3′3′-cGAMP cleavage activity by thin-layer chromatography. Acb1 was identified as the product of the phage T4 gene *57B*.

## Phage genome sequencing, assembly and annotation of SBSphiJ1–7

SBSphiJ1–7 phages were isolated from soil samples on *B. subtilis* BEST7003 culture as described previously[33]. High-titre phage lysates (>$10^7$ PFUs ml$^{-1}$) were used for DNA extraction. A 500 μl volume of the phage lysate was treated with DNase-I (Merck catalogue number 11284932001) added to a final concentration of 20 mg ml$^{-1}$ and incubated at 37 °C for 1 h to remove bacterial DNA. DNA was extracted using the QIAGEN DNeasy blood and tissue kit (catalogue number 69504) starting from the Proteinase-K treatment step to lyse the phages. Libraries were prepared for Illumina sequencing using a modified Nextera protocol as previously described[38]. Following Illumina sequencing, adapter sequences were removed from the reads using Cutadapt version 2.8 (ref. [39]) with the option -q 5. The trimmed reads from each phage genome were assembled into scaffolds using SPAdes genome assembler version 3.14.0 (ref. [40]), using the --careful flag. Each assembled genome was analysed with Prodigal version 2.6.3 (ref. [41]; default parameters) to predict open reading frames.

## SBSphiJ Apyc1 bioinformatic identification

The genomic sequences of SBSphiJ and the closely related family members SBSphiJ1–7 were aligned using progressive Mauve (ref. [42]). Regions that were exclusive to cCMP-cleaving phages revealed eight candidate genes. The corresponding SBSphiJ protein sequences were analysed using HHpred (ref. [43]) for predicted structural homologues. Protein classes with >75% probability are listed in Extended Data Fig. 5b and Apyc1 was identified as the product of the phage SBSphiJ gene *147*.

## Identification of Acb1 and Apyc1 homologues and generation of phylogenetic trees

Homologues of Acb1 and Apyc1 were identified using NCBI BLASTp with default parameters. Acb1 sequences were classified as belonging to a prophage if they were within three genes of a phage structural or packaging protein. Apyc1 phage sequences were identified by restricting the search to only viral sequences (NCBI taxid: 10293; https://www.ncbi.nlm.nih.gov/Taxonomy/Browser/wwwtax.cgi?id=10239). Maximum-likelihood trees were generated using the IQ-TREE web server with ultrafast bootstrapping and 1,000 iterations[44]. Consensus trees were then edited visually using the Interactive Tree Of Life[45].

## Crystallization and structure determination

Crystals were grown in hanging-drop format using EasyXtal 15-well trays (NeXtal). Crystals of native and selenomethionine-labelled phage FBB1 Acb1 G8–D152 were grown at 18 °C in 2-μl drops mixed 1:1 with purified protein (4 mg ml$^{-1}$, 20 mM HEPES-KOH pH 7.5, 80 mM KCl, 1 mM TCEP) and reservoir solution (2 M ammonium sulfate, 0.1 M sodium citrate pH 4.6). Crystals were grown for 1–7 days before being cryo-protected with reservoir solution supplemented with 45% sucrose and collected by freezing in liquid nitrogen. Crystals of the FBB1 Acb1–3′3′-cGAMP complex were grown using the same reservoir conditions, except drops and cryo-protectant solution were supplemented with 100 μM of a hydrolysis-resistant phosphorothioate-modified analogue of 3′3′-cGAMP (Biolog Life Science Institute, C 216). Crystals of Bsp38 Apyc1 were grown at 18 °C in 2-μl drops mixed 1:1 with purified protein (10 mg ml$^{-1}$, 20 mM HEPES-KOH pH 7.5, 80 mM KCl, 1 mM TCEP) and reservoir solution (0.2 M lithium sulfate, 0.1 M Tris-HCl pH 7.5, 30% PEG-4000). Crystals were grown for 1–7 days before being cryo-protected with reservoir solution supplemented with 15% glycerol and collected by freezing in liquid nitrogen. Crystals of selenomethionine-labelled *Paenibacillus J14* (GenBank accession number WP_028539944.1) Apyc1 were grown at 18 °C in 2-μl drops mixed 1:1 with purified protein (10 mg ml$^{-1}$, 20 mM HEPES-KOH pH 7.5, 80 mM KCl, 1 mM TCEP) and reservoir solution (0.1 M Tris-HCl pH 8.5, 0.2 M MgCl$_2$, 16% PEG-4000) supplemented with 100 μM of a hydrolysis-resistant phosphorothioate-modified analogue of cAMP (Biolog Life Science Institute, A 003). Crystals were grown for 1–7 days before being cryo-protected with reservoir solution supplemented with 25% ethylene glycol and collected by freezing in liquid nitrogen. Crystals of *Paenibacillus xerothermodurans* Apyc1 were grown at 18 °C in 2-μl drops mixed 1:1 with purified protein (10 mg ml$^{-1}$, 20 mM HEPES-KOH pH 7.5, 80 mM KCl, 1 mM TCEP) and reservoir (0.1 M HEPES-KOH pH 7.5, 0.2 M calcium acetate, 10% PEG-8000). Crystals were grown for 1–7 days before being cryo-protected with reservoir solution supplemented with 25% ethylene glycol and collected by freezing in liquid nitrogen. X-ray diffraction data were collected at the Advanced Photon Source (beamlines 24-ID-C and 24-ID-E), and data were processed using the SSRL autoxds script (A. Gonzalez, Stanford SSRL). For Acb1 and Apyc1 phase determination, anomalous data were collected using selenomethionine-labelled Acb1 crystals, heavy sites were identified with HySS in Phenix (ref. [46]), and an initial map was produced using SOLVE/RESOLVE in Phenix (ref. [46]). Model building was performed using Coot (ref. [47]), and then refined in Phenix. Statistics were analysed as described in Supplementary Table 2 (refs. [48–50]). Final structures were refined to stereochemistry statistics for Ramachandran plot (favoured/allowed), rotamer outliers and MolProbity score as follows: FBB1 Acb1, 98.52%/1.48%, 0.8% and 1.11; FBB1 Acb1–3′3′-cGAMP, 99.26%/0.74%, 1.56% and 1.39; Bsp38 Apyc1, 90.79%/7.46%, 4.85% and 2.64; *P. J14* Apyc1, 95.04%/4.96%, 2.38% and 1.78; *P. xerothermodurans* Apyc1, 96.12%/3.88%, 1.93% and 1.60. See Supplementary Table 2 and the Data availability statement for the deposited PDB codes. All structure figures were generated with PyMOL 2.3.0.

## HPLC

Acb1 and Apyc1 reactions for HPLC analysis were performed in a 100 μl volume and consisted of 50 mM Tris-HCl pH 7.5, 100 mM KCl, 1 mM DTT, 100 μM chemically synthesized nucleotide standards (Biolog Life Science Institute) and 1 μM recombinant protein unless otherwise indicated. Apyc1 reactions were further supplemented with 5 mM MgCl$_2$ and 1 mM MnCl$_2$. Reactions were incubated at 37 °C for 20 min (unless otherwise indicated in the figure legend) and filtered using a 10-kDa cutoff filter (Millipore). Filtered nucleotide products were analysed using a C18 column (Agilent Zorbax Bonus-RP 4.6 × 150 mm, 3.5 μm) heated to 40 °C and run at 1 ml min$^{-1}$ in a buffer of 50 mM NaH$_2$PO$_4$ adjusted to pH 6.8 with NaOH, supplemented with 3% acetonitrile.

## In vitro reconstitution of CBASS and Pycsar effector function and inhibition

Synthetic cyclic nucleotides (Biolog Life Science Institute) were pre-incubated with purified T4 Acb1 and SBSphiJ Apyc1 in reactions containing 1 μM cyclic nucleotide, 1 μM recombinant Acb1 or Apyc1 protein, 50 mM Tris-HCl pH 7.5, 100 mM KCl and 1 mM DTT for 1 h at 37 °C. Apyc1 reactions were further supplemented with 5 mM $MgCl_2$ and 1 mM $MnCl_2$. Cyclic nucleotide reactions were then used as $10\times$ inputs for effector activation reactions using the following recombinant CBASS and Pycsar effector proteins: *V. cholerae* CapV (ref. [11]), *E. cloacae* Cap4 (ref. [23]), *B. pseudomallei* Cap5 (ref. [23]) and *B. cepacia* PycTIR (ref. [2]). Nuclease effectors were incubated in 25-μl reactions containing 1 μM effector protein, and buffer consisting of 50 mM Tris-HCl pH 7.5, 25 mM NaCl, 5 mM $MgCl_2$, 1 mM DTT and 10 ng $μl^{-1}$ pGEM9z plasmid DNA. Following 20-min incubation at 37 °C, 5 μl of DNA loading dye was added and 15 μl was analysed on a 1% agarose gel as previously described[23]. CapV phospholipase activity was analysed in 25-μl reactions consisting of 1 μM purified effector, 50 mM Tris-HCl pH 7.5, 25 mM NaCl, 5 mM $MgCl_2$, 1 mM DTT and a BODIPY-labelled EnzChek phospholipase substrate (ThermoFisher) as previously described[11]. Phospholipase activity was measured using a Synergy H1 plate reader (BioTek) according to the manufacturer's instructions. PycTIR was used at 40 μM in 25-μl reactions consisting of 20 mM HEPES-KOH pH 7.5, 100 mM KCl and 500 μM of the fluorescent $NAD^+$ analogue ε-NAD (Sigma). Fluorescent measurements (300 nm excitation, 410 nm emission) were taken in a Synergy H1 plate reader (BioTek) following 2-min incubation at room temperature.

## Bacterial growth assays

CBASS effector function was measured in *E. coli* using conditions that result in autoactivation of *V. cholerae* DncV 3′3′-cGAMP synthesis as previously described[36]. *E. coli* BL21(DE3) competent cells (NEB) were transformed with three plasmids encoding *V. cholerae* CapV (pBAD33), the CBASS effector *B. pseudomallei* Cap5 (pET16), and either WT or catalytically inactive (H44A/H113A) T4 Acb1 (pTU175)[51]. Transformations were plated onto MDG plates and three colonies were picked and grown for 16 h (37 °C, 230 r.p.m.) in 5-ml MDG starter cultures. A 5 ml volume of M9ZB cultures was inoculated with 200 μl MDG starter culture and grown for 3 h (37 °C, 230 r.p.m.) before being induced by diluting 1:5 in M9ZB media containing 5 μM IPTG and 0.2% L-arabinose. Induced culture (200 μl) was added to wells of a 96-well plate, and $OD_{600}$ was read every 6.82 min for 300 min in a Synergy H1 plate reader (BioTek) while shaking at 230 r.p.m., 37 °C. Wells containing medium alone were used for $OD_{600}$ background subtraction.

## Phage challenge assays

Phage challenge experiments were performed as previously described[1,2] by spotting serial dilutions of high-titre phage stocks onto a lawn of bacteria carrying a complete CBASS or Pycsar defence operon. The following defence systems were used: *E. coli* strain KTE188 (IMG gene accession numbers: 2564596481–2564596485; https://img.jgi.doe.gov/) cloned under its native promoter into the plasmid pSG1 (ref. [3]), *E. coli* CdnG cloned under its native promoter into the plasmid pLOCO2 (ref. [23]), *Y. aleksiciae* CdnE (ref. [36]) cloned into a pBAD vector, and *E. coli* PycC (ref. [2]) cloned under its native promoter into the plasmid pSG1. For *Ec*CdnG and *Ya*CdnE operons, control plasmids were also used in which the CD-NTase is inactivated (CdnG-D82A/D84A)[23] or the transmembrane segment of the receptor is deleted (*Ya*CdnE)[36]. Phage replication in the context of these defence systems was measured using a spot plaque assay[36]. Briefly, *E. coli* MG1655 (*Ec*KTE188, *Ec*PycC) or BL21 cells (*Ec*CdnG and *Ya*CdnE) containing the defence systems were grown overnight at 37 °C. A 300 μl volume of the bacterial culture was mixed with 4 ml melted MMB agar containing appropriate antibiotics and 0.2% arabinose for pBAD plasmids, poured on top of a 15-cm plate of

lysogeny broth and left to solidify in a plate for 1 h at room temperature. High-titre phage stocks were serially diluted tenfold in MMB and 3–5-μl drops were placed on the bacterial layer and allowed to dry at room temperature for 1 h. Plates were incubated overnight at 37 °C (Acb1 and Apyc1 rescue experiments) or 30 °C (Δ*acb1* T4 phage challenges) and plaque-forming units (PFUs) were determined by counting the derived plaques after overnight incubation. Phage infection of cells expressing active CBASS operons did not generate clear plaques. For these, the dilution at which there was no detectable defect in bacterial growth was counted as having a single plaque. For in vivo rescue experiments, *acb1* and *apyc1* were amplified from the genome of T4 phage or SBSphiJ phage and cloned into the plasmid pBbS8k (Addgene number 35276) using Gibson assembly (NEB).

## Generation of phage T4 ΔAcb1

Nonsense mutations were introduced into *acb1* using a CRISPR-based selection strategy as described previously[52,53]. Briefly, a gRNA targeting *acb1* and a repair template with nonsense mutations were cloned into pCRISPR (Addgene 42875). *E. coli* Top10 cells were then transformed with the pCRISPR–gRNA-*acb1* repair plasmid and pCas9 (Addgene 42876). A colony was picked, and 2-ml log-scale cultures were infected with WT phage T4 until culture collapse. The resulting lysate was filtered through a 0.22-μM filter and plated on *E. coli* Top10 cells with no plasmid. Single plaques were picked into 200 μl SM buffer (50 mM Tris-HCl pH 8.5, 100 mM NaCl, 8 mM $MgSO_4$) containing 2 μl chloroform. After 1-h incubation at room temperature, 4 μl was used as input for standard PCR reactions using GoTaqGreen (Promega) according to the manufacturer's instructions. PCR products were purified using QIAquick gel extraction kit (Qiagen) and sequenced for introduction of nonsense mutations. Positive phage T4 clones went through three rounds of plaque purification before generating a high-titre stock used in all phage challenge experiments.

## Statistics and reproducibility

Statistical tests are described in the figure legends and were performed using GraphPad Prism 9.3.1. Experimental details regarding replicates and sample size are described in the figure legends. No statistical methods were used to predetermine sample size and no blinding or randomization was used for this study.

## Reporting summary

Further information on research design is available in the Nature Research Reporting Summary linked to this paper.

## Data availability

Coordinates and structure factors of FBB1 Acb1, the FBB1 Acb1–3′3′-cGAMP complex, Bsp38 Apyc1, *P. J14* Apyc1 and *P. xerothermodurans* Apyc1 have been deposited in the PDB under the accession codes 7T26, 7T27, 7T28, 7U2R and 7U2S, respectively. Source data are provided with this paper. All other data are available in the manuscript or the Supplementary Information.

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

**Acknowledgements** We are grateful to B. Duncan-Lowey, and members of the laboratory of P.J.K. for helpful comments and discussion. Mass spectrometry was performed at the Biopolymers and Proteomics Core Facility at the Koch Institute of the Massachusetts Institute of Technology and the Taplin Mass Spectrometry Facility at Harvard Medical School. The work was financed by grants to P.J.K. from the Pew Biomedical Scholars programme, the Burroughs Wellcome Fund PATH programme, The Mathers Foundation, The Mark Foundation for Cancer Research, the Parker Institute for Cancer Immunotherapy and the National Institutes of Health (1DP2GM146250-01) and grants to R.S. from the European Research Council (ERC-AdG GA 101018520), the Israel Science Foundation (ISF 296/21), the Ernest and Bonnie Beutler Research Program of Excellence in Genomic Medicine, the Deutsche Forschungsgemeinschaft (SPP 2330, grant 464312965), the Minerva Foundation and the Knell Family Center for Microbiology. S.J.H. is supported through a Cancer Research Institute Irvington Postdoctoral Fellowship (CRI3996), T.W. is supported through a Minerva Foundation postdoctoral fellowship and a European Molecular Biology Organization postdoctoral fellowship (ALTF 946-2020), and B.R.M. is supported as a Ruth L. Kirschstein National Research Service Award Postdoctoral Fellow (NIH F32GM133063). X-ray data were collected at the Northeastern Collaborative Access Team beamlines 24-ID-C and 24-ID-E (P30 GM124165), and used a Pilatus detector (S10RR029205), an Eiger detector (S10OD021527) and the Argonne National Laboratory Advanced Photon Source (DE-AC02-06CH11357).

**Author contributions** Experiments were designed and conceived by S.J.H. and P.J.K. Anti-CBASS and anti-Pycsar biochemical screen was performed by S.J.H. with phage-infected samples prepared by T.W. Phage Acb1 and Apyc1 identification and validation experiments were performed by S.J.H. Crystallography and biochemical experiments were performed by S.J.H. and A. Lu, with assistance from J.S. NADase effector assays were performed by B.R.M. Phage genome sequencing was performed by A. Leavitt and E.Y. Phage challenge assays were performed by T.W., S.J.H. and R.S. The manuscript was written by S.J.H. and P.J.K. All authors contributed to editing the manuscript and support the conclusions.

**Competing interests** R.S. is a scientific cofounder and advisor of BiomX and Ecophage. The other authors declare no competing interests.

**Additional information**
**Correspondence and requests for materials** should be addressed to Philip J. Kranzusch.

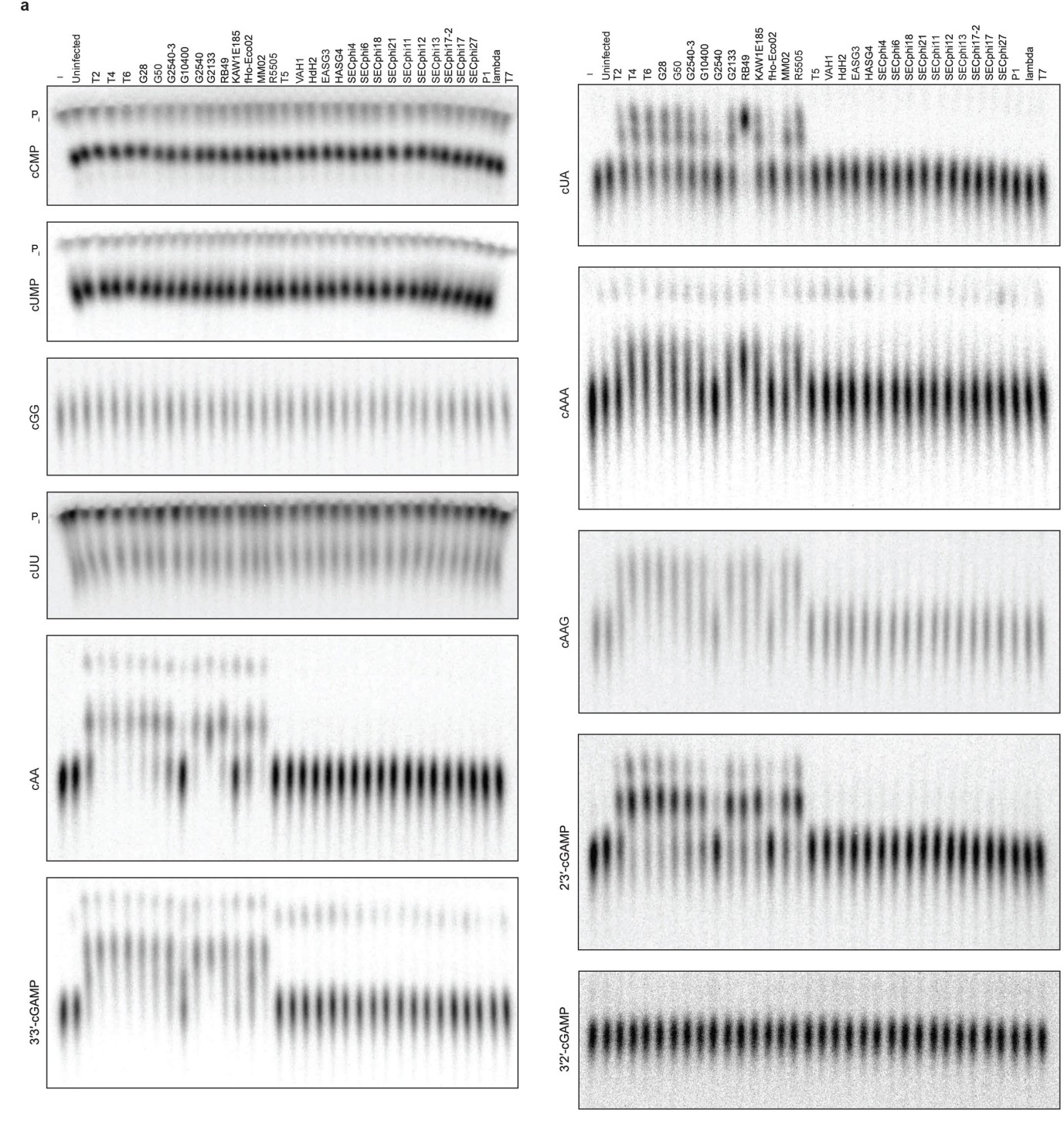

**Extended Data Fig. 1 | A biochemical screen to discover coliphage anti-CBASS and anti-Pycsar evasion. a**, Representative TLC assays depicting coliphage degradation of radiolabeled cyclic nucleotides after 1 h incubation in infected *E. coli* lysates.

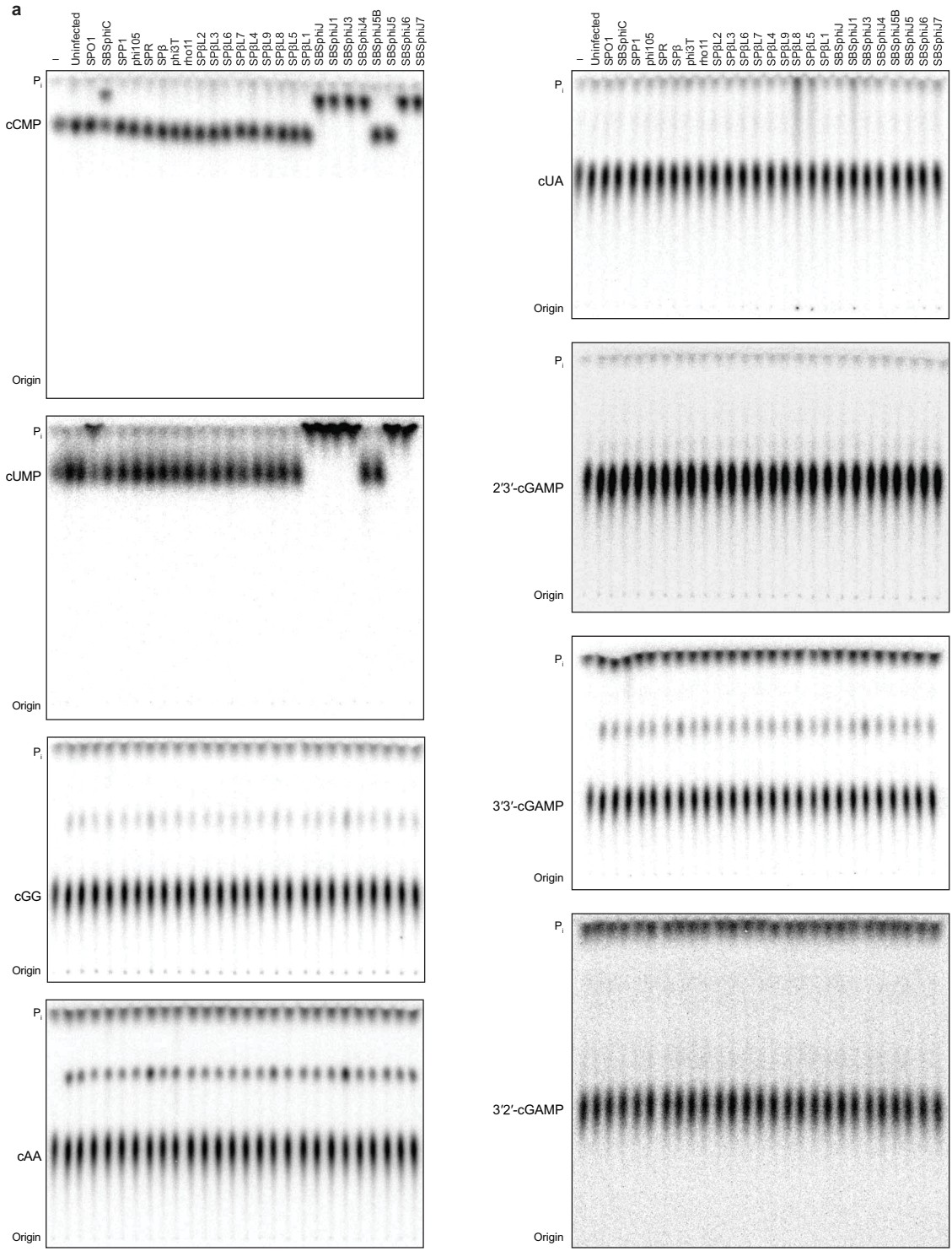

**Extended Data Fig. 2 | A biochemical screen to discover *Bacillus* phage anti-CBASS and anti-Pycsar evasion. a**, Representative TLC assays depicting *Bacillus* phage degradation of radiolabeled cyclic nucleotides after 1 h incubation in infected *B. subtilis* lysates.

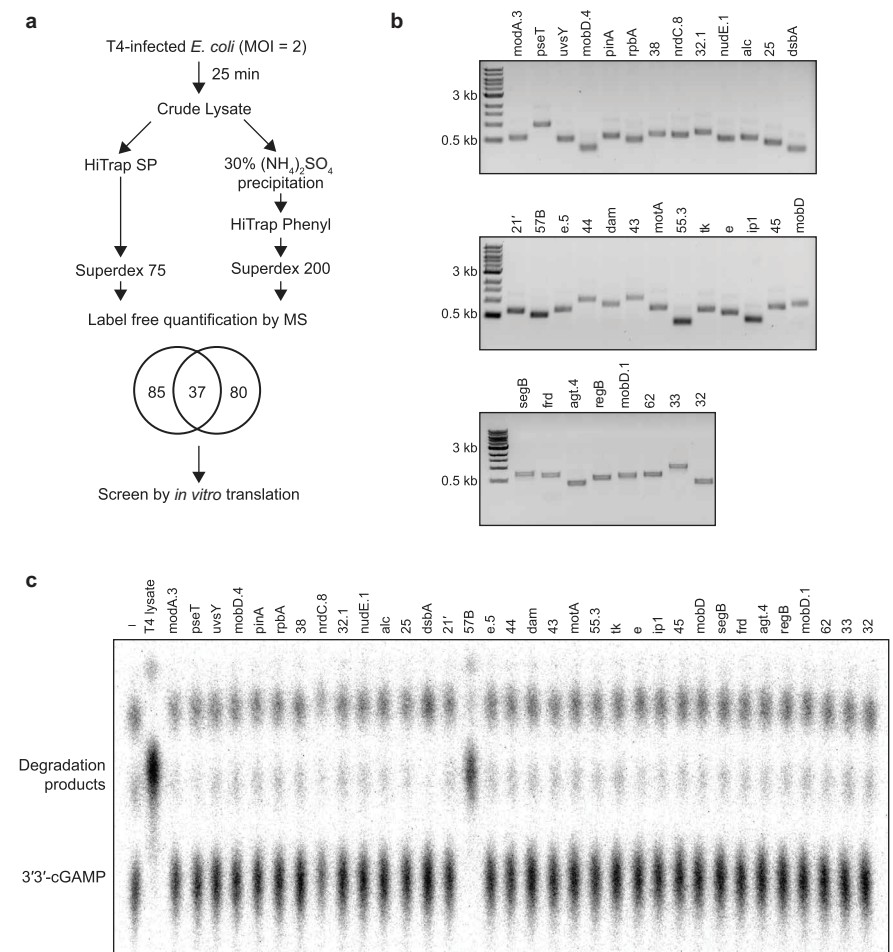

**Extended Data Fig. 3 | Biochemical fractionation and mass spectrometry identification of phage T4 gene *57B* as Acb1. a**, Schematic depicting strategies of biochemical fractionation to enrich 3′3′-cGAMP cleavage activity from crude T4 lysate. Mass spectrometry of active fractions revealed 37 candidate T4 proteins. **b**, Agarose gel analysis of 34 successfully PCR-amplified candidate T4 genes to be screened for 3′3′-cGAMP cleavage activity by *in vitro* translation. Data are representative of 2 independent experiments. For gel source data, see Supplementary Figure 1. **c**, Translation products from b were tested for 3′3′-cGAMP cleavage activity by TLC. Data are representative of 2 independent experiments.

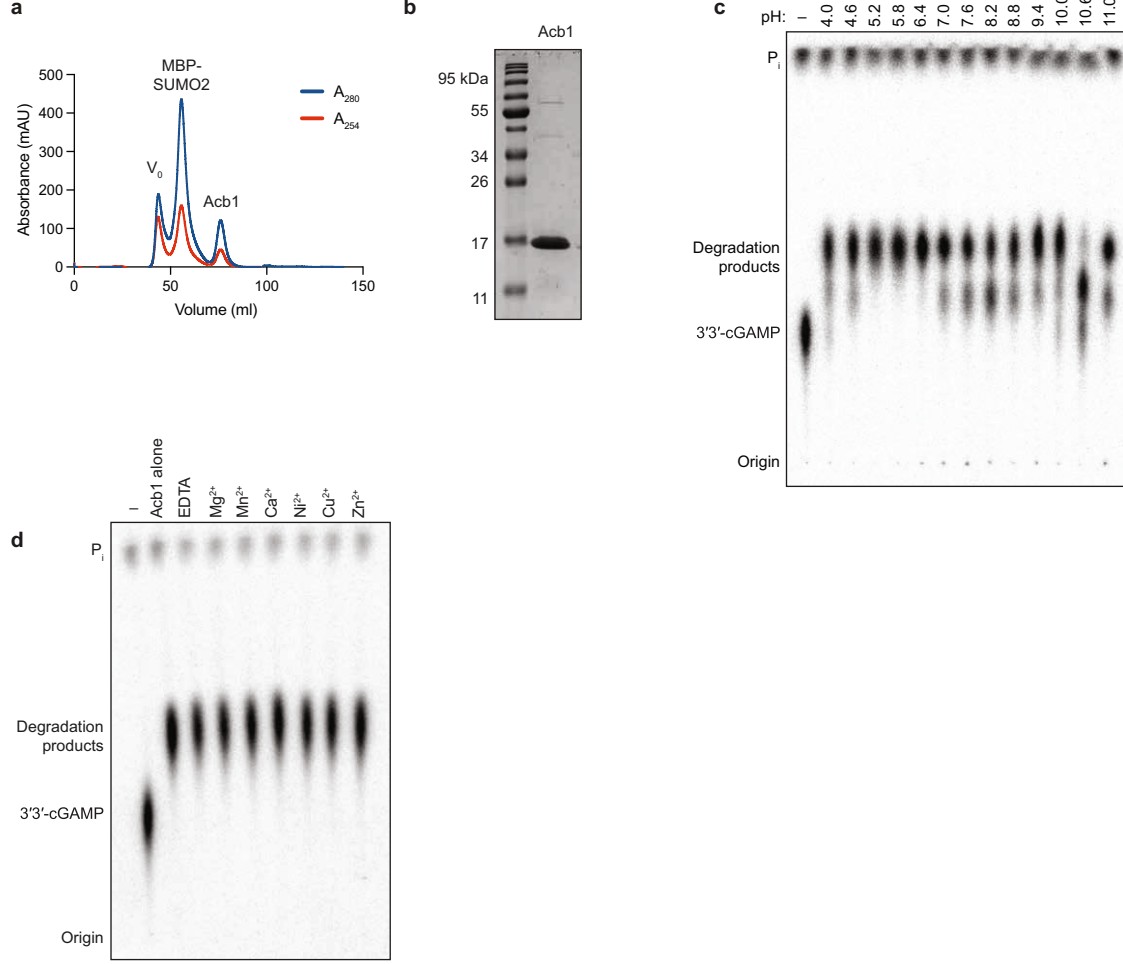

**Extended Data Fig. 4 | Purification and biochemical characterization of recombinant T4 Acb1. a**, T4 Acb1 was expressed as an N-terminal 6×His-MBP-SUMO2 fusion and purified by Ni-NTA and separated from His-MBP-SUMO2 by size exclusion chromatography. **b**, Coomassie-stained SDS-PAGE analysis of fully purified T4 Acb1. **c**, TLC analysis of 3′3′-cGAMP degradation by T4 Acb1 at the indicated pH. **d**, TLC analysis of 3′3′-cGAMP degradation by T4 Acb1 supplemented with the indicated metal or EDTA at the following concentrations: 50 mM EDTA; 5 mM MgCl$_2$; 1 mM MnCl$_2$; 5 mM CaCl$_2$; 1 μM NiCl$_2$; 1 μM CuCl$_2$; or 1 μM ZnSO$_4$. Data in all panels are representative of at least 3 independent experiments.

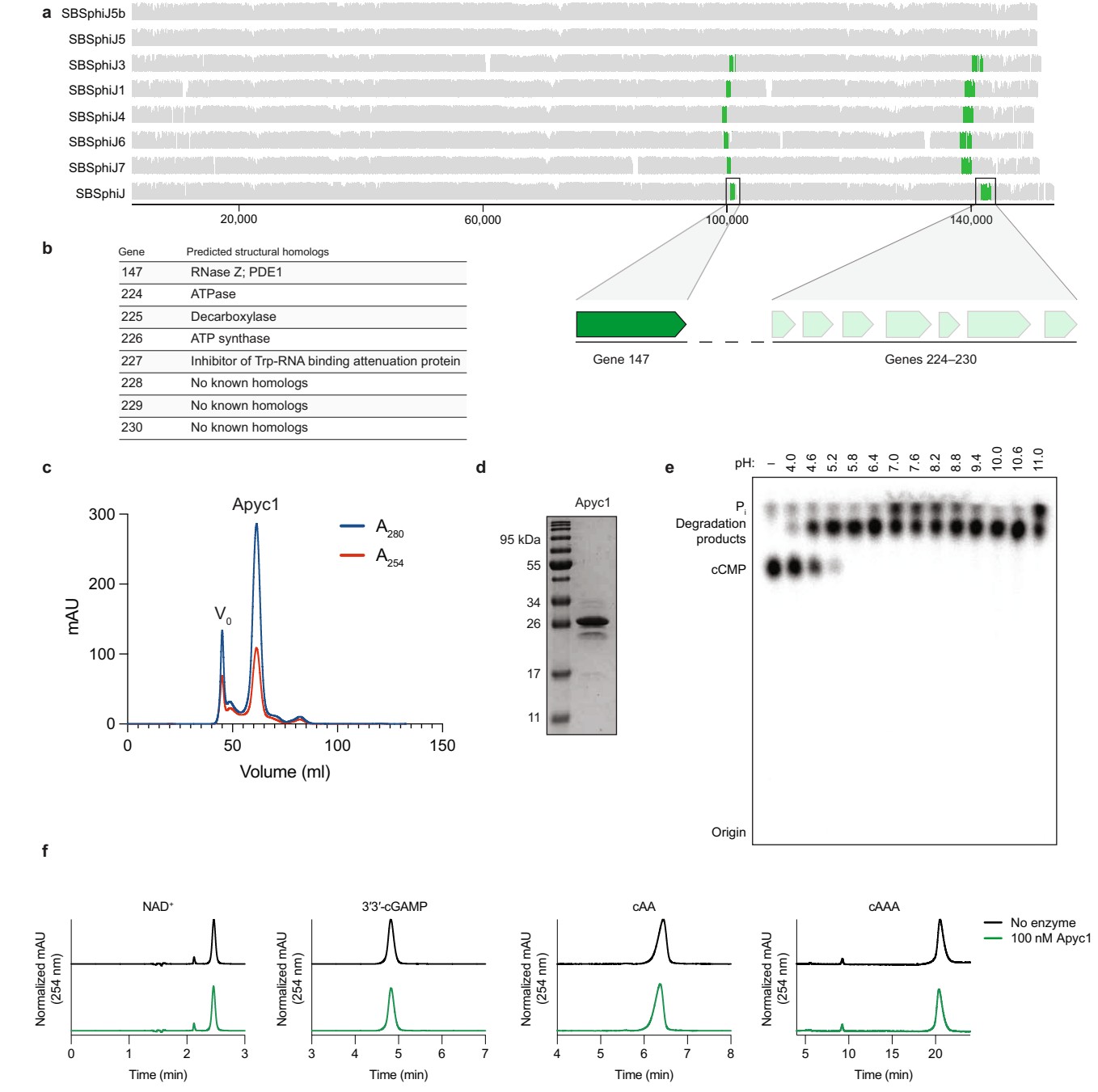

**Extended Data Fig. 5 | Bioinformatic identification and biochemical characterization of phage SBSphiJ gene *147* as the anti-Pycsar nuclease Apyc1. a**, Genome schematic of SBSphiJ and 7 other closely related phages highlighting regions exclusive to cCMP-cleaving phages. **b**, Summary of HHpred analysis of candidate genes. **c**, Recombinant SBSphiJ Apyc1 was expressed as an N-terminal 6×His-SUMO2 fusion, purified by Ni-NTA, and separated from His-SUMO2 by size exclusion chromatography. **d**, Coomassie-stained SDS-PAGE analysis of fully purified SBSphiJ Apyc1. **e**, TLC analysis of cCMP degradation by SBSphiJ Apyc1 at the indicated pH. **f**, HPLC analysis of recombinant SBSphiJ Apyc1 incubated with the indicated substrates (100 μM) for 30 min at 37 °C. Data in c–f are representative of 3 independent experiments.

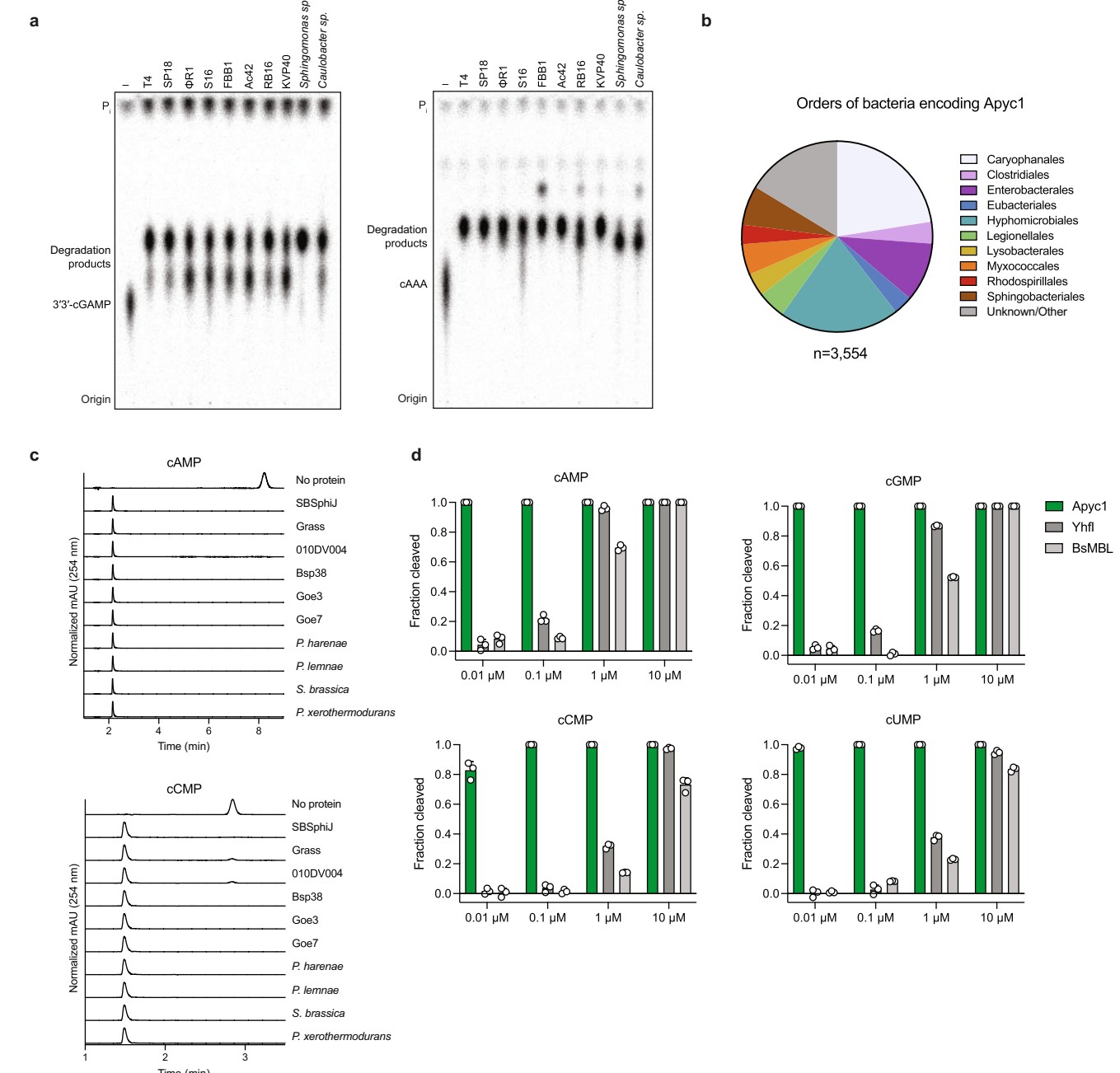

**Extended Data Fig. 6 | Substrate specificity of host and viral enzymes related to Acb1 and Apyc1. a**, Homologs of T4 Acb1 were expressed, purified, and tested for cleavage of 3′3′-cGAMP (left) and cAAA (right) by TLC. Data are representative of 2 independent experiments. **b**, Summary of the distribution of bacterial Apyc1 homologs among bacterial orders. **c**, Homologs of SBSphiJ Apyc1 were expressed, purified, and tested for cleavage of cAMP and cCMP by HPLC. Data are representative of 3 independent experiments. **d**, Summary of HPLC analysis of cNMP degradation by SBSphiJ Apyc1 and closely related *B. subtilis* MBL phosphodiesterases. Data are presented as mean ± s.d. from n = 3 independent replicates.

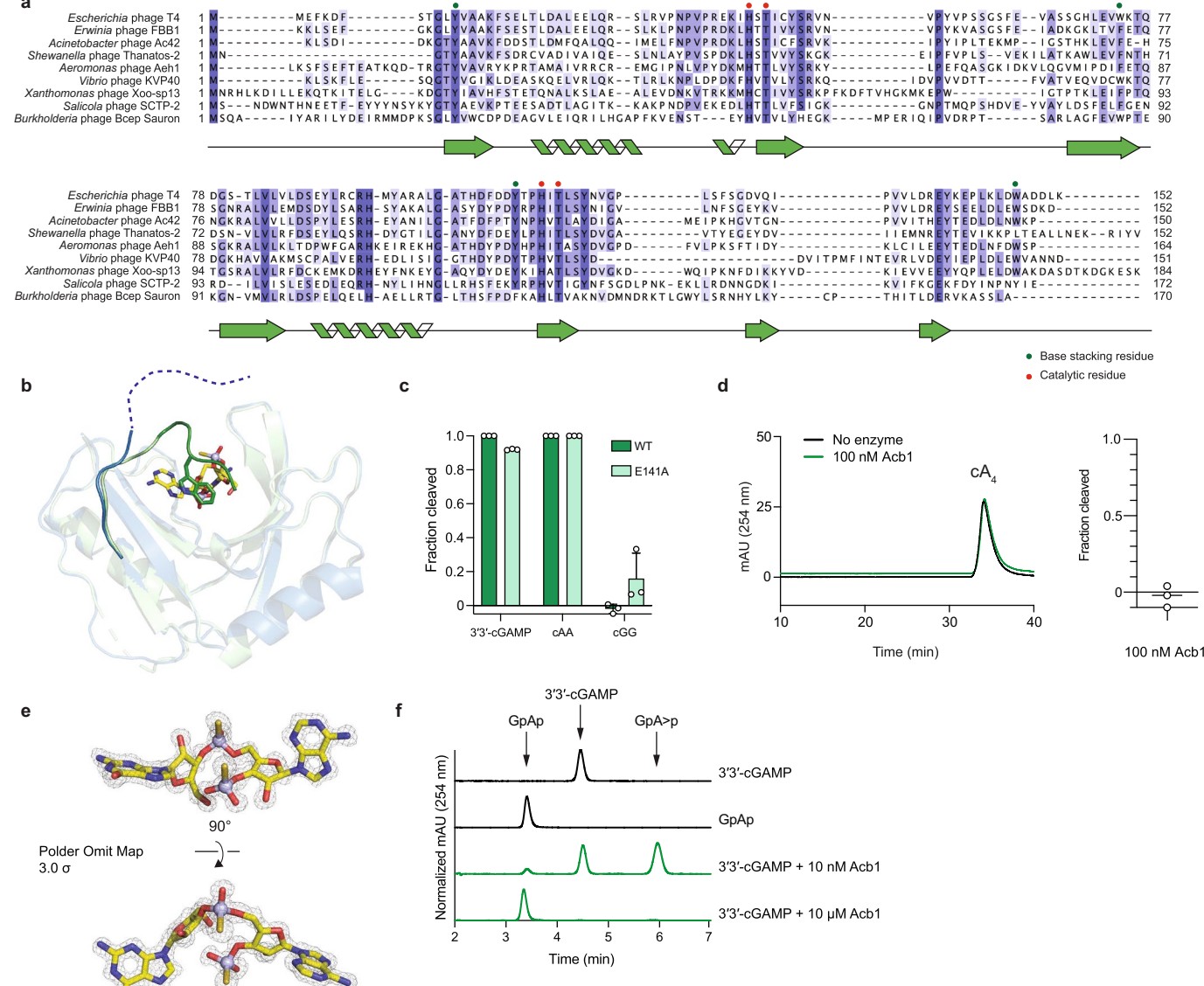

**Extended Data Fig. 7 | Structural analysis of Acb1 and mechanism of 3′3′-cGAMP cleavage. a**, Structure guided multiple sequence alignment of Acb1 proteins from the indicated phages. The strength of shading indicates degree of residue conservation. **b**, Overview of Acb1 in the *apo* state (blue) and bound to 3′3′-cGAMP (green). The C-terminal lid is unstructured in the apo state and encloses 3′3′-cGAMP upon binding. **c**, Summary of HPLC analysis of WT or E141A T4 Acb1 cleavage of the indicated substrate. Data are presented as

mean ± s.d. from n = 3 independent experiments. **d**, HPLC analysis of cA₄ cleavage by T4 Acb1. Data in graph are presented as mean ± s.d. from n = 3 independent experiments. **e**, Polder omit map of 3′3′-cGAMP contoured at 3.0 σ. **f**, Comparison of T4 Acb1 3′3′-cGAMP degradation products and synthetic 3′3′-cGAMP and GpAp standards by HPLC. Data are representative of 2 independent experiments.

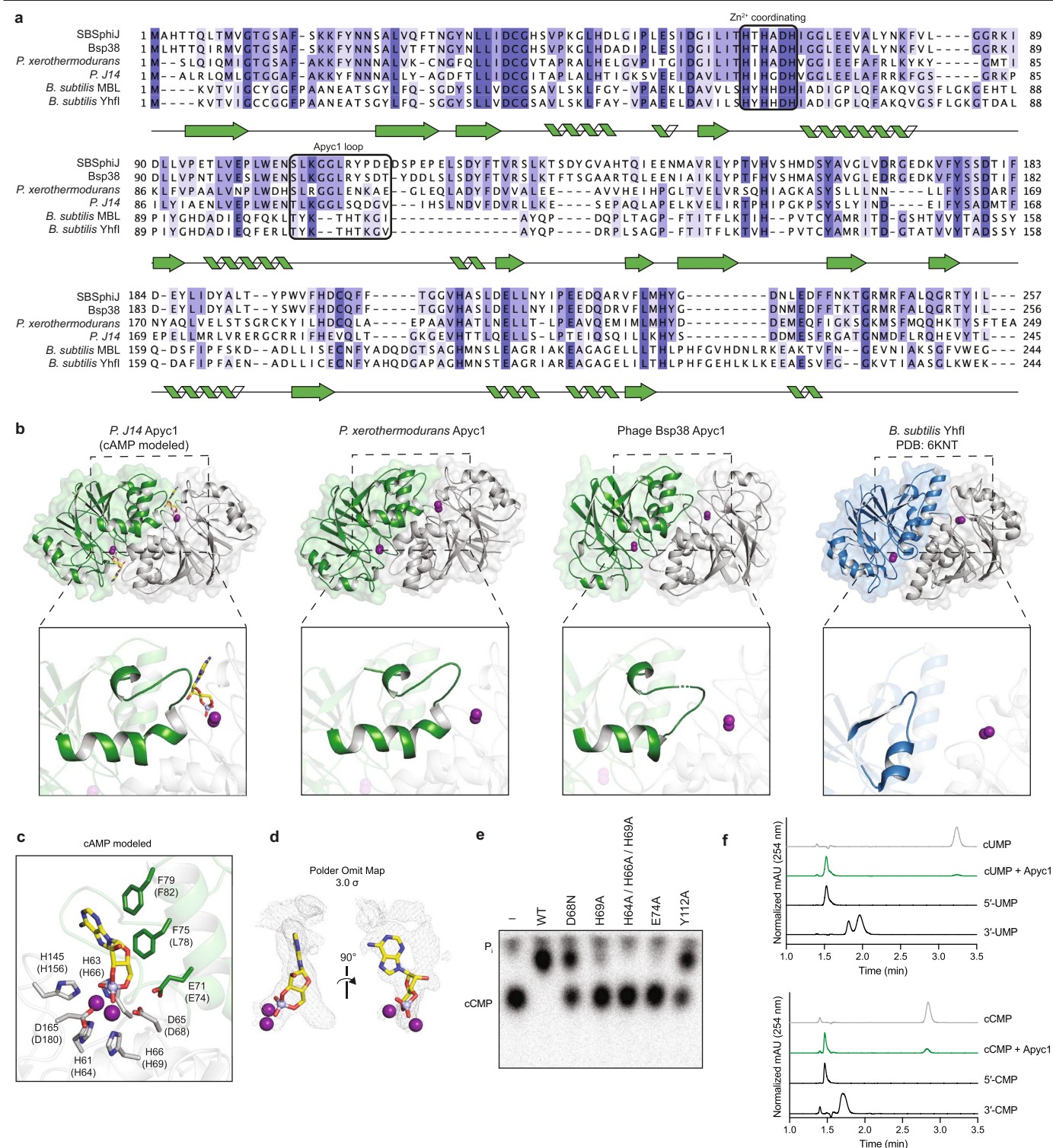

**Extended Data Fig. 8 | Structural analysis of Apyc1 and mechanism of cNMP degradation. a**, Structure guided multiple sequence alignment of Apyc1 proteins from the indicated phages or bacterial species and *B. subtilis* MBL phosphodiesterases. The strength of shading indicates degree of residue conservation. **b**, Overview of *P. J14* Apyc1, *P. xerothermodurans* Apyc1, Bsp38 Apyc1, and *B. subtilis* YhfI crystal structures, with one monomer in colour and one monomer in grey. Detailed area highlights an Apyc1-specific loop that extends into the cNMP binding pocket. **c**, Detailed view of the residues coordinating the $Zn^{2+}$ ions with cAMP modeled into the cNMP binding pocket.

Numbers in parentheses indicate equivalent residue in SBSphiJ Apyc1. **d**, *P. J14* Apyc1 crystallized in the presence of a hydrolysis-resistant phosphorothioate analog of cAMP resulted in clear phosphate and ribose density in the binding pocket and sparse density corresponding to the nucleobase. Polder omit map of cAMP contoured at 3.0 σ. **e**, TLC analysis of cCMP cleavage by SBSphiJ Apyc1 point mutants. **f**, HPLC analysis of Apyc1 cCMP and cUMP reaction products compared to synthesized 5′-CMP, 5′-UMP and 3′-CMP or 3′-UMP standards. Data are representative of 2 independent experiments.

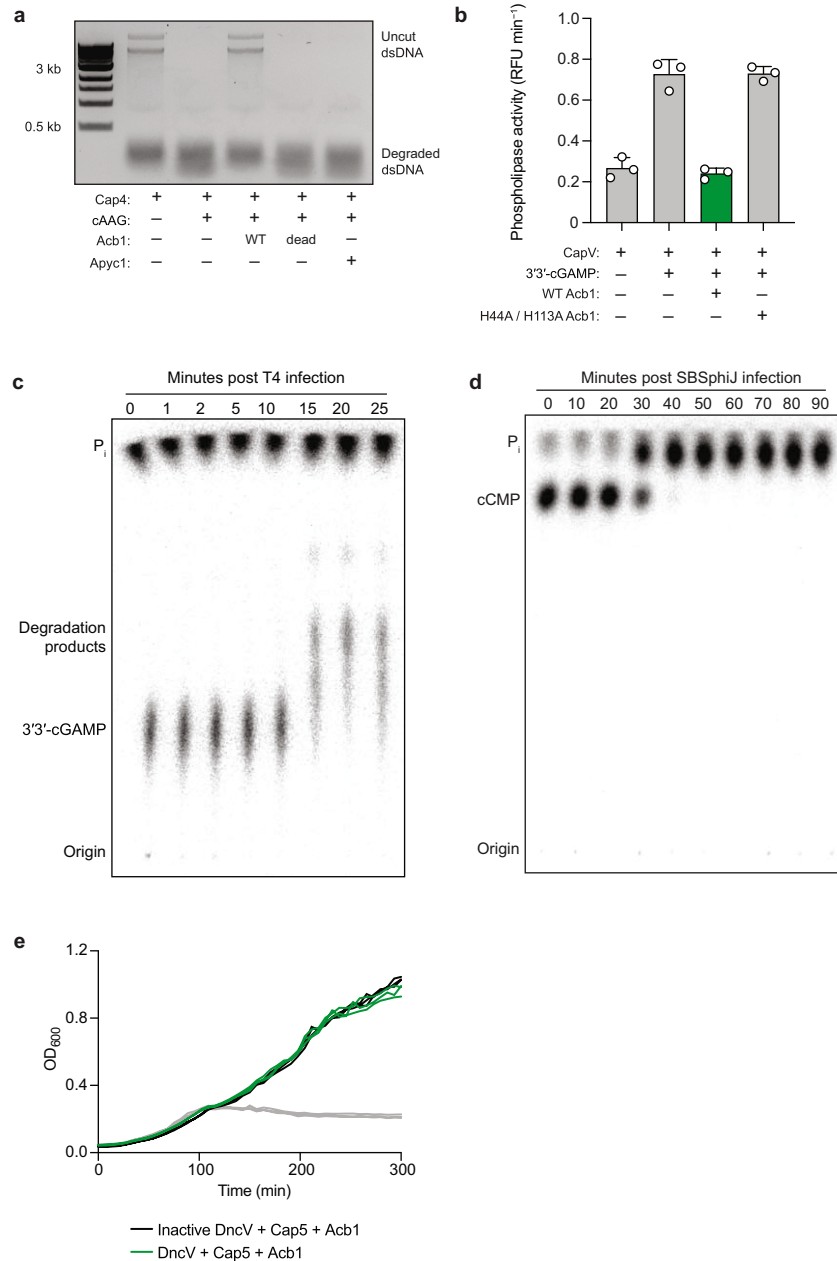

**Extended Data Fig. 9 | Effector inhibition and time course analysis of Acb1 and Apyc1 activity. a**, Agarose gel analysis of uncut plasmid DNA incubated with Cap4 and cAAG that was treated with WT Acb1, catalytically inactive Acb1 H44A, or WT Apyc1. Data are representative of 3 independent experiments. For gel source data, see Supplementary Figure 1. **b**, Release of fluorescent dye from a phospholipid substrate incubated with recombinant CapV and 3′3′-cGAMP that was treated with WT or catalytically inactive H44A/H113A Acb1. Data are presented as mean ± s.d. from n = 3 independent experiments. **c**, T4-infected cells were collected at the indicated time point, and lysates were tested for 3′3′-cGAMP cleavage activity by TLC. Data are representative of 3 independent experiments. **d**, SBSphiJ-infected cells were collected at the indicated time point and lysates were tested for cCMP cleavage activity. Data are representative of 2 independent experiments. **e**, Bacterial growth in cells expressing Cap5, WT or D132A/D134A catalytically inactive DncV, and WT or H44A/H113A catalytically inactive T4 Acb1. Technical replicates are plotted and the data are representative of 3 independent experiments.

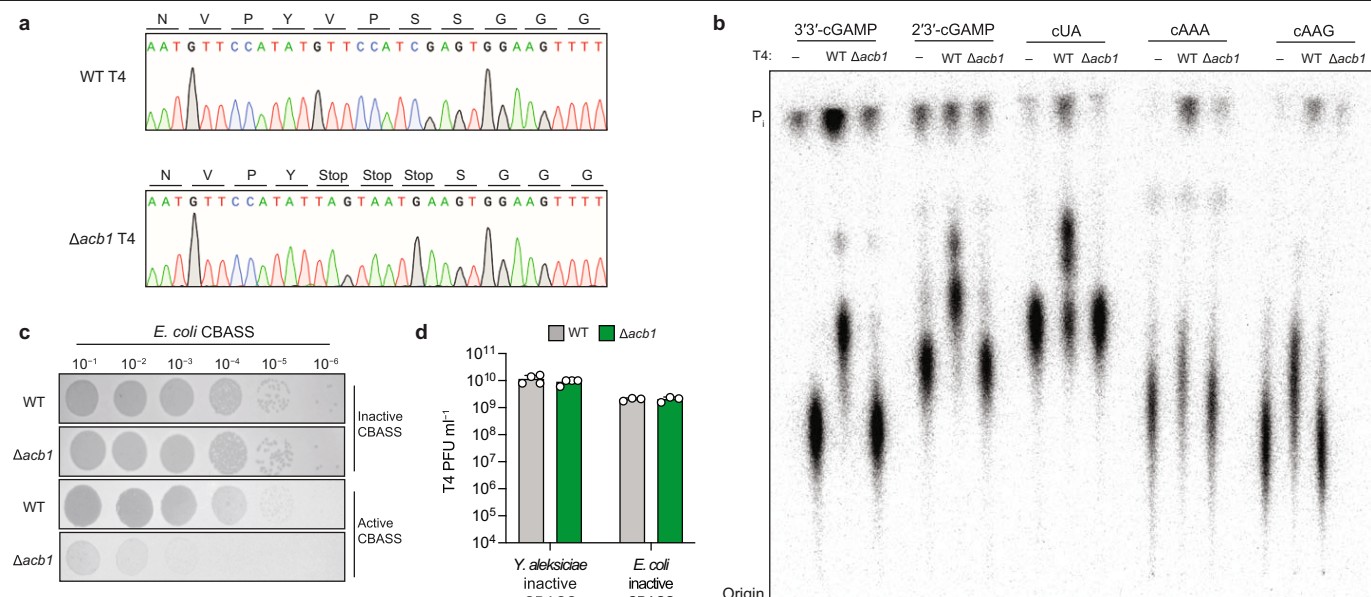

**a**, Sequencing reads of WT and Δacb1 phage T4. **b**, TLC analysis of cyclic nucleotide cleavage by WT or Δacb1 phage T4 lysate. Data are representative of 2 independent experiments. **c**, Representative plaque assays of *E. coli* carrying a plasmid encoding an active or catalytically inactive CBASS operon from

**Extended Data Fig. 10 | Generation and validation of phage T4 Δacb1.**

*Y. aleksiciae* **d**, Summary of plaque assay results of WT or Δacb1 phage T4 infection of *E. coli* carrying catalytically inactive CBASS operons from *Y. aleksiciae* or *E. coli*. Data are presented as mean ± s.d. from n = 4 (*Y. aleksiciae* operon) or n = 3 technical replicates (*E. coli* operon) and are representative of at least 3 biologically independent experiments.

| | |
|---|---|

# Reporting Summary

## Statistics

For all statistical analyses, confirm that the following items are present in the figure legend, table legend, main text, or Methods section.

| n/a | Confirmed | |
|---|---|---|
| ☐ | ☒ | The exact sample size (*n*) for each experimental group/condition, given as a discrete number and unit of measurement |
| ☐ | ☒ | A statement on whether measurements were taken from distinct samples or whether the same sample was measured repeatedly |
| ☐ | ☒ | The statistical test(s) used AND whether they are one- or two-sided<br>*Only common tests should be described solely by name; describe more complex techniques in the Methods section.* |
| ☒ | ☐ | A description of all covariates tested |
| ☒ | ☐ | A description of any assumptions or corrections, such as tests of normality and adjustment for multiple comparisons |
| ☒ | ☐ | A full description of the statistical parameters including central tendency (e.g. means) or other basic estimates (e.g. regression coefficient) AND variation (e.g. standard deviation) or associated estimates of uncertainty (e.g. confidence intervals) |
| ☐ | ☒ | For null hypothesis testing, the test statistic (e.g. *F*, *t*, *r*) with confidence intervals, effect sizes, degrees of freedom and *P* value noted<br>*Give P values as exact values whenever suitable.* |
| ☒ | ☐ | For Bayesian analysis, information on the choice of priors and Markov chain Monte Carlo settings |
| ☒ | ☐ | For hierarchical and complex designs, identification of the appropriate level for tests and full reporting of outcomes |
| ☒ | ☐ | Estimates of effect sizes (e.g. Cohen's *d*, Pearson's *r*), indicating how they were calculated |

*Our web collection on statistics for biologists contains articles on many of the points above.*

## Software and code

Policy information about availability of computer code

| Data collection | Protein homologs were identified using NCBI BLAST.<br>Phylogenetic trees were constructed using the IQ-TREE web server v1.6.12.<br>Radiographic images were collected using Typhoon scanner control 2.0.0.6<br>Chromatography traces were collected using GE Unicorn 7.1<br>DNA gel images were collected using BioRad Quantity One 4.6.9 |
|---|---|
| Data analysis | Phenix 1.19, Coot 0.8.9, PyMOL 2.3, Prism 9.3.1, iTOL v6 |

For manuscripts utilizing custom algorithms or software that are central to the research but not yet described in published literature, software must be made available to editors and reviewers. We strongly encourage code deposition in a community repository (e.g. GitHub). See the Nature Portfolio guidelines for submitting code & software for further information.

## Data

Policy information about availability of data

All manuscripts must include a data availability statement. This statement should provide the following information, where applicable:
- Accession codes, unique identifiers, or web links for publicly available datasets
- A description of any restrictions on data availability
- For clinical datasets or third party data, please ensure that the statement adheres to our policy

Coordinates and structure factors of FBB1 Acb1, the FBB1 Acb1–3'3'-cGAMP complex, Bsp38 Apyc1, Paenibacillus J14 Apyc1, and Paenibacillus xerothermodurans Apyc1 have been deposited in PDB under the accession codes 7T26, 7T27, 7T28, 7U2R, and 7U2S, respectively. All other data are available in the manuscript or the supplementary materials.

# Field-specific reporting

Please select the one below that is the best fit for your research. If you are not sure, read the appropriate sections before making your selection.

☒ Life sciences  ☐ Behavioural & social sciences  ☐ Ecological, evolutionary & environmental sciences

For a reference copy of the document with all sections, see nature.com/documents/nr-reporting-summary-flat.pdf

# Life sciences study design

All studies must disclose on these points even when the disclosure is negative.

| | |
|---|---|
| Sample size | Samples sizes were chosen based on previously published protocols (Lowey et al, 2020, PMID: 32544385). |
| Data exclusions | No data were excluded from analysis. |
| Replication | All experimental findings were successfully replicated multiple times. |
| Randomization | X-ray crystal structures were refined using a randomly selected set of R-free reflections. |
| Blinding | Blinding was not required in this study as data were collected using highly quantitative measures over multiple independent replicates. |

# Reporting for specific materials, systems and methods

We require information from authors about some types of materials, experimental systems and methods used in many studies. Here, indicate whether each material, system or method listed is relevant to your study. If you are not sure if a list item applies to your research, read the appropriate section before selecting a response.

## Materials & experimental systems

| n/a | Involved in the study |
|---|---|
| ☒ ☐ | Antibodies |
| ☒ ☐ | Eukaryotic cell lines |
| ☒ ☐ | Palaeontology and archaeology |
| ☒ ☐ | Animals and other organisms |
| ☒ ☐ | Human research participants |
| ☒ ☐ | Clinical data |
| ☒ ☐ | Dual use research of concern |

## Methods

| n/a | Involved in the study |
|---|---|
| ☒ ☐ | ChIP-seq |
| ☒ ☐ | Flow cytometry |
| ☒ ☐ | MRI-based neuroimaging |

