## [Peer Review File · Nature]

Manuscript Title: Phage anti-CBASS and anti-Pycsar nucleases subvert bacterial immunity

Reviewer Comments & Author Rebuttals

Reviewer Reports on the Initial Version:

Referees' comments:

Referee #1 (Remarks to the Author):

This manuscript investigates two families of phage-encoded phosphodiesterases that degrade cyclic nucleotides involved in anti-phage signaling. The authors present the following new findings:

1. A protein family called Acb1 is encoded in T4-like phages. These proteins degrade cyclic nucleotides produced by CBASS anti-phage systems. They do not degrade cAMP or cGMP, which are involved in bacterial signaling pathways.
2. A protein family called Apyc1. These proteins degrade cUMP and cCMP, signaling molecules involved in another anti-phage system called Pycsar. These proteins also degrade cAMP and cGMP and are described as non-specific phosphodiesterases.
3. The crystal structures of Acb1 and Apyc1 were determined. These provided insight into how these enzymes function and showed that their mechanisms and structures are unrelated.
4. Various in vivo experiments showed that both Acb1 and Apyc1 proteins block activity of the CBASS and Pycsar systems, respectively. When these systems were used to target phages and prevent their replication, co-expression of Acb1 or Apyc1 inhibited the anti-phage systems allowing phage replication.

The manuscript is high quality, presenting a thorough and extensive set of experiments. The results are certainly exciting and interesting. In particular, the approach used to discover Acb1 and Apyc1 was clever and could be applied to other systems that signal through specific cyclic nucleotides. This paper should be published in a good journal. However, there is a deficiency that, I believe, makes it fall short for publication in Nature, which I describe below.

The authors state at the end of the paper: "Overall, our results define viral nucleases as a key mechanism of CBASS and Pycsar immune evasion and reveal the role of viral proteins in driving evolution of cyclic nucleotide-based immune defense systems"; and in the abstract: "Here we show that phages encode anti-CBASS (Acb) and anti-Pycsar (Apyc) proteins that counteract defense by specifically degrading cyclic nucleotide signals that activate host immunity". The problem is that while the authors have shown that Acb1 and Apyc1 can block the named anti-phage systems, they have not proven that this is the function of these proteins within the context of a phage. In other words, they have not shown that any phage relies on these proteins for replication in a strain that is expressing CBASS or Pycsar. The missing experiment is, for example, to delete the *acb1* gene from a phage and show that this phage can no longer replicate due to being blocked by a CBASS system. Or

they could add the *acb1* gene to a phage that is blocked by CBASS (e.g. T5) and show that the phage is no longer blocked. In the case of *Apyc1*, the authors describe very similar phages, some of which possess *Apyc1* and others that do not. If there were a *Bacillus subtilis* strain with an active Pycsar system then showing that only the phages with *Apyc1* can replicate on this strain would also be a good experiment. The issue I raise here may seem semantic but saying that these proteins mediate immune evasion by phages is simply wrong unless data are presented to prove it.

The problem alluded to above is particularly acute in the case of *Apyc1*. *Apyc1* cleaves cAMP and cGMP, which are involved in many bacterial processes, as well as cCMP and cUMP. Phage proteins are known to interfere with bacterial signaling processes, such as quorum sensing. Thus, it is equally possible that the function of *Apyc1* is to interfere with cAMP or cGMP mediated signaling. There are also many more bacterial homologues than phage homologues. This would not be expected for a protein involved in immune evasion (or are these bacterial homologues in prophages?). Is Pycsar a commonly occurring anti-phage system in *Bacillus*? Are *Apyc1* homologs found in species that have Pycsar systems? Some correlation would be expected. The authors argue for the uniqueness of the *Apyc1* proteins by comparing to another phosphodiesterase of the same family (MBL). They say, “MBL phosphodiesterase exhibited a strong preference for cAMP/cGMP over cCMP/cUMP cleavage, confirming relaxed nucleotide specificity and Pycsar signal degradation is unique to *Apyc1* and not a general feature of MBL phosphodiesterase enzymes (Extended Data Figure 6e)”. This is an overstatement. In Extended Data Figure 6f, we see that there is only a 4-fold difference in activity of MBL phosphodiesterase against cGMP and cUMP. So, this enzyme still has a considerable degree of non-specificity. This one example does not convince me that relaxed specificity is a unique feature of *Apyc1*.

Some minor issues:

Line 86 needs to be re-worded.

In Fig. 1c, why does the *B. subtilis* extract degrade 3'3'3'-cAAA and 3'3'3'-cAAG, and the phage infected extracts do not?

I would call “*Apyc1*” “*Apy1*” to be consistent with typical bacterial gene/protein naming (3 letters and a number) and to be consistent with anti-CRISPR (*Acr*) and anti-CBASS (*Acb*).

Referee #2 (Remarks to the Author):

This manuscript describes the discovery of virus encoded inhibitors for the recently discovered CBASS and Pycsar phage defense systems. Although virus encoded nucleases for signalling molecules have been found before against other virus defense systems (CRISPR, cGAS-STING), this is an interesting and important manuscript that helps to further understand CBASS and Pycsar, and its escape by phage. Although the paper is clearly written, the assembly and order of results is somewhat unusual because of two systems investigated here, and the broad combination of different techniques used, making the paper somewhat unfocused. Still the data is of high quality

and the insights obtained are broad. Two points need to be addressed.

Major points

- It is unclear if the phages that naturally carry the inhibitor genes can overcome CBASS and Pycsar immunity. Do they escape immunity and to what extent?
- Further, in order to demonstrate that phage use *Acb1* and *Apyc1* genes to evade CBASS and Pycsar defense systems it would be necessary to knock-out (or knock-down) the inhibitor genes in the natural phage. Now only the introduction of *Acb1* and *Apyc1* is shown in trans and this is tested with phage P1 and T5 that naturally do not contain these inhibitors. The suggested experiment will determine if these inhibitors actually contribute to a phenotype.

Referee #3 (Remarks to the Author):

In their manuscript, Hobbs et al. describe the function, structure and mechanisms of phage-encoded proteins that allow phages to evade CBASS and Pycsar immunity.

First, the authors incubate phage-infected culture lysates with a large array of cyclic nucleotides that could be potential signalling molecules involved in prokaryotic immunity. This shows that various phages encode proteins that are able to degrade one or more (types of) signalling molecules. Next, they identify and two suspect enzymes, *Acb1* and *Apyc1*, and show the phylogeny of these enzymes which are encoded in a plethora of phages. Furthermore, it is demonstrated that these enzymes can degrade different types of signalling molecules in vitro, confirming they are the enzymes responsible for degradation of the signalling molecules. Next, the authors have solved the structures of apo-*Acb1*, substrate-bound *Acb1*, and apo-*Apyc1*. Structure-guided mutants are made and provide insights into the catalytic mechanism and substrate specificity of these enzymes. Finally, the authors demonstrate both in vivo and in vitro that *Acb1* (but not *Apyc1*) can prevent *Cap5* activity in presence of 3'3'-cGAMP, while *Apyc1* (but not *Acb1*) can prevent *PycTIR* activity in presence of cUMP, implying their activity specifically prevents activation of each immune system.

The methodology is appropriate, the data supports the novel and exciting conclusions drawn, and the manuscript extremely well written. I only have a couple of suggestions.

1. The authors present *Apyc1* and *Acb1* as anti-Pycsar and anti-CBASS enzymes in their manuscript. However, for their function, these enzymes might not have to specifically interact with Pycsar and CBASS-system proteins (as certain antiCRISPR proteins do). Have the authors investigated if there are interactions between these immune systems and the characterized proteins? *Apyc1* and *Acb1* degrade messenger molecules that theoretically could also be used by other (to-be-characterized) prokaryotic immune systems. For example, certain type III CRISPR systems are known to (also) generate cA3 molecules recognized by *Acb1* ((Gruschow et al 2019, Kazlauskienė et al, 2017, Han et al 2018). While most characterized effectors rely on other cA# molecules for their activation, it is also known that many putative effectors exist, and they might be regulated by such signalling molecules. Therefore, it might well be that *Apyc1* and *Acb1* also interfere with other immune

systems. The authors should at least reflect on this.

2. Almost all in vitro assays and in vivo experiments lack a negative control, preferentially a sample in which a catalytic mutant of the enzyme is used. This should at least be added for the in vivo experiments (to confirm the observed effects are a consequence of Acb1/Apyc1 activity, and not just because of their expression). Otherwise, it cannot be ruled out that these proteins can inhibit the systems in another way (e.g. by direct binding).

4. It should be discussed in the introduction that *E. coli* strains naturally encode Pycsar and CBASS systems. This adds to the significance that *E. coli* phages encode inhibitors of this system. Are the systems also found in *Bacillus subtilis* strains? This should be discussed, also when this is not the case.

5. The R values for the Apyc1 structure are (too) high for a 2.7Å structure. Furthermore, there is a high amount of Ramachandran and sidechain outliers. This structure should be refined further, or an explanation should be given for the poor statistics. Once the structure has been further refined, I would recommend to add the structure-function characterization of Apyc1 to Figure 3 too, as it is currently underrepresented.

6. The authors claim that Apyc1 has a broad specificity, but only show it degrades cNMPs (in contrast to Acb1, for which they show degradation of different types of cyclic nucleotides). This suggests its activity is selective for very similar cNMPs (and not broad as claimed) - it is just not selective for specific bases on cNMPs. Does Apyc1 also degrade other types of (cyclic nucleotide) substrates? Based on Figure 1 I would guess this is not the case, and I would rephrase the claim to something like 'it degrades cNMPs with no selection for specific bases'.

7. It is unclear what the error bars in Figure 2 and 4, and several extended data figures indicate.

Referee #4 (Remarks to the Author):

The authors of this manuscript identify and characterize a novel class of enzymes that specifically counter cyclic nucleotide based anti-phage defense systems. Starting from the observation that recently identified anti-phage signaling compounds like 3'3'-cGAMP or cCMP are highly stable in bacterial cell extracts, but rapidly degraded when challenged with extracts of phage-infected cells, they set out to biochemically define the respective hydrolytic activities. This led to the discovery of novel, phage-encoded hydrolytic enzymes which display broad recognition of different cyclic nucleotides and which confer the ability to overrule nucleotide-based phage defense in vitro and in vivo. Based on these findings and on the observation that members of these enzyme families are widespread in phylogenetically diverse phages, the authors conclude that degradation of host-derived cyclic nucleotides represents a common immune evasion strategy of phages.

This work is clearly exciting and novel and of broad relevance as it adds additional components of phage immune evasion thereby contributing to a better understanding of bacterial innate immunity. Experiments are well executed and convincing. I only have a few inquiries and suggestions.

It is not clear why Apyc1 was discovered specifically with phages infecting *B. subtilis*. The authors recently demonstrated that *E. coli* strains do encode Pycsar immunity components. However, none

of the E. coli phages seems to be equipped with nucleases targeting Pycsar. Please comment.

Fig. 2i,j The figures display the distribution of Acb1 and Apyc1 homologs without providing specific information about the respective phylogeny. It would be valuable for the reader to have some additional information to understand how broadly these enzymes are distributed and if most (all?) of them are associated with prophages.

Fig. 3 The specificity of Acb1 for cyclic nucleotides containing at least one adenine moiety is intriguing. It is also interesting that specificity for A is mediated by the same residue (E141) that rotates into the active site upon substrate binding, apparently inflicting a rotation of the adenine base and a stretched substrate conformation. Is it possible that the observed limited substrate specificity of Acb1 is dictated by the catalytic mechanism of these enzymes? Or in other words, do substrates require at least one A and specific base contact because a strained substrate conformation is critical for catalysis? The observation that the E141A mutant variant has retained activity against cGAMP *in vitro*, argues that this is not the case, but no kinetic data are presented. It would be very interesting to investigate if the E141A mutant no longer discriminates against non-As like cGG.

In the last paragraph of the manuscript the authors speculate that diversification of cyclic nucleotide signals is a key host adaptation to maintain anti-phage defense (l.166) and that viral nucleases have a prominent role in driving the evolution of cyclic nucleotide mediated immune defense systems (l.181). This appears to make little sense, given the very broad substrate specificities of phage-encoded hydrolytic activities. Please specify.

l.177 'The astounding diversity...'. This statement is unclear. Please clarify.

Author Rebuttals to Initial Comments:

Referee #1

This manuscript investigates two families of phage-encoded phosphodiesterases that degrade cyclic nucleotides involved in anti-phage signaling. The authors present the following new findings:

- 1. A protein family called Acb1 is encoded in T4-like phages. These proteins degrade cyclic nucleotides produced by CBASS anti-phage systems. They do not degrade cAMP or cGMP, which are involved in bacterial signaling pathways.*
- 2. A protein family called Apyc1. These proteins degrade cUMP and cCMP, signaling molecules involved in another anti-phage system called Pycsar. These proteins also degrade cAMP and cGMP and are described as non-specific phosphodiesterases.*
- 3. The crystal structures of Acb1 and Apyc1 were determined. These provided insight into how these enzymes function and showed that their mechanisms and structures are unrelated.*
- 4. Various in vivo experiments showed that both Acb1 and Apyc1 proteins block activity of the CBASS and Pycsar systems, respectively. When these systems were used to target phages and prevent their replication, co-expression of Acb1 or Apyc1 inhibited the anti-phage systems allowing phage replication.*

The manuscript is high quality, presenting a thorough and extensive set of experiments. The results are certainly exciting and interesting. In particular, the approach used to discover Acb1 and Apyc1 was clever and could be applied to other systems that signal through specific cyclic nucleotides. This paper should be published in a good journal. However, there is a deficiency that, I believe, makes it fall short for publication in Nature, which I describe below.

We thank the reviewer for finding our manuscript exciting and of high quality. We are grateful for their helpful feedback and have focused our revisions on new biochemistry, structural biology, and virology experiments to further define the role of cyclic nucleotide degrading enzymes in phage evasion of bacterial immune defense.

The authors state at the end of the paper: "Overall, our results define viral nucleases as a key mechanism of CBASS and Pycsar immune evasion and reveal the role of viral proteins in driving evolution of cyclic nucleotide-based immune defense systems"; and in the abstract: "Here we show that phages encode anti-CBASS (Acb) and anti-Pycsar (Apyc) proteins that counteract defense by specifically degrading cyclic nucleotide signals that activate host immunity". The problem is that while the authors have shown that Acb1 and Apyc1 can block the named anti-phage systems, they have not proven that this is the function of these proteins within the context of a phage. In other words, they have not shown that any phage relies on these proteins for replication in a strain that is expressing CBASS or Pycsar. The missing experiment is, for example, to delete the acb1 gene from a phage and show that this phage can no longer replicate due to being blocked by a CBASS system. Or they could add the acb1 gene to a phage that is blocked by CBASS (e.g. T5) and show that the phage is no longer blocked. In the case of Apyc1, the authors describe very similar phages, some of which possess Apyc1 and others that do not. If there were a Bacillus subtilis strain with an active Pycsar system then showing that only the phages with Apyc1 can replicate on this strain would also be a good experiment. The issue I raise here may seem semantic but saying that these proteins mediate immune evasion by phages is simply wrong unless data are presented to prove it.

We thank the reviewer for this suggestion. To address the reviewer's point, we focused on the well-characterized phage T4 and performed extensive new experiments to define the *in vivo* importance of cyclic nucleotide-specific nucleases in immune evasion. First, we adapted a recently developed CRISPR/Cas9-based strategy (Tao et al, 2017 PMID 28657724) to introduce nonsense mutations and engineer a mutant phage T4 Δ Acb1 virus (Extended Data Figure 10a). *E. coli* cells infected with phage T4 Δ Acb1 exhibit no ability to hydrolyze 3'3'-cGAMP or other cyclic nucleotides, confirming that Acb1 is essential for viral degradation of CBASS immune signals (Extended Data Figure 10b). We next compared growth of wildtype phage T4 and phage T4 Δ Acb1 in *E. coli* encoding functional or catalytically inactive CBASS immune systems. In the absence of functional CBASS defense, phage T4 and phage T4 Δ Acb1 grow equally well and release 10^{11} particles mL⁻¹ revealing that Acb1 is not required for normal replication in *E. coli*. In contrast, growth of phage T4 Δ Acb1 is specifically impaired in the presence of active CBASS immunity with the mutant virus exhibiting a >1,000 fold defect in viral replication compared to wildtype phage T4 (Figure 4e,f; Extended Data Figure 10c,d). Combined with our previous experiments showing Acb1 expression rescues growth of phage P1 from CBASS-

mediated inhibition (Figure 4c), these results demonstrate that viral nucleases are necessary and sufficient for evasion of cyclic nucleotide-mediated phage defense.

The problem alluded to above is particularly acute in the case of Apyc1. Apyc1 cleaves cAMP and cGMP, which are involved in many bacterial processes, as well as cCMP and cUMP. Phage proteins are known to interfere with bacterial signaling processes, such as quorum sensing. Thus, it is equally possible that the function of Apyc1 is to interfere with cAMP or cGMP mediated signaling. There are also many more bacterial homologues than phage homologues. This would not be expected for a protein involved in immune evasion (or are these bacterial homologues in prophages?). Is Pycsar a commonly occurring anti-phage system in Bacillus? Are Apyc1 homologs found in species that have Pycsar systems? Some correlation would be expected. The authors argue for the uniqueness of the Apyc1 proteins by comparing to another phosphodiesterase of the same family (MBL). They say, "MBL phosphodiesterase exhibited a strong preference for cAMP/cGMP over cCMP/cUMP cleavage, confirming relaxed nucleotide specificity and Pycsar signal degradation is unique to Apyc1 and not a general feature of MBL phosphodiesterase enzymes (Extended Data Figure 6e)". This is an overstatement. In Extended Data Figure 6f, we see that there is only a 4-fold difference in activity of MBL phosphodiesterase against cGMP and cUMP. So, this enzyme still has a considerable degree of non-specificity. This one example does not convince me that relaxed specificity is a unique feature of Apyc1.

Our data demonstrate that Apyc1 hydrolyzes cNMP nucleotides independent of nucleobase identity. In contrast, all previously characterized cNMP hydrolyases exhibit preferential specificity for cAMP/cGMP nucleotides. To address the reviewer's point and further support our characterization of Apyc1, our revised manuscript now includes a series of new biochemical experiments and two new structures including a co-crystal structure of *Paenibacillus* Apyc1 in complex with an analog of cAMP.

First, to further confirm the unique ability of Apyc1 to efficiently degrade cCMP/cUMP, we purified *B. subtilis* Yhfl as an additional example MBL-family bacterial cNMP phosphodiesterase and tested this enzyme's ability to target the specialized Pycsar nucleotides cCMP and cUMP. These new results confirm Yhfl and conventional MBL-family enzymes have a dramatically lower ability to hydrolyze cCMP/cUMP compared to viral Apyc1 proteins. Apyc1 is >100-fold more active on cCMP/cUMP substrates compared to MBL-family phosphodiesterases, with 10 nM enzyme sufficient to turn-over 100 μ M of substrate cCMP or cUMP in a 20 min reaction (Extended Data Figure 6d). Additionally, we tested alternative nucleotide substrates and confirmed that Apyc1 has no ability to degrade non-cNMP substrates including 3'3'-cGAMP, 3'3'-cAA, 3'3'3'-cAAA and NAD⁺ (Extended Data Figure 5f). Second, we have performed in depth bioinformatic analysis of Apyc1 and now provide an updated phylogeny that includes >100 viral enzymes demonstrating that Apyc1 is widely conserved within phages (Figure 2j). Closely related Apyc1 enzymes are also found present in multiple, diverse classes of bacteria (Extended Data Figure 6b). To further define the relationship between phage and bacterial Apyc1 proteins, we determined a 1.5 Å crystal structure of a *Paenibacillus* Apyc1 enzyme in the apo state and a 1.8 Å crystal structure of a *Paenibacillus* Apyc1 enzyme in complex with an analog of cAMP (Extended Data Figure 8b–d). The new *Paenibacillus* structures confirm close homology with phage Bsp38 Apyc1 and conservation of an open active-site architecture and an extended Apyc1-specific active-site loop that support cNMP hydrolysis independent of nucleobase identity (Extended Data Figure 8b–f). Together, our new results further confirm that both phage and *Paenibacillus* Apyc1 enzymes rapidly degrade cCMP/cUMP and exhibit activity that is unique compared to other previously characterized cNMP phosphodiesterases.

Finally, although Apyc1 is capable of degrading cAMP and cGMP, our gain-of-function experiments with phage T5 demonstrate that expression of Apyc1 is well-tolerated in *E. coli* and does not interfere with phage T5 replication (Figure 4d). While Apyc1 may have additional accessory functions targeting other nucleotide signals, all our results support that this previously uncharacterized protein enables evasion of Pycsar immune defense. As the technology necessary for *B. subtilis* phage engineering is poorly developed, we note in the text that our knock-out studies focus on phage T4 (see Lines 173–174). Extending understanding of the specific role of Apyc1 in phage SBSphiJ anti-Pycsar evasion is a goal for future experiments.

Some minor issues:

Line 86 needs to be re-worded.

We have revised the text as follows: “However, *Apyc1* is encoded adjacent to a series of small proteins of unknown function, suggesting that this variable loci in SBSphiJ-family phages may contribute to evasion of other anti-phage defense systems.” (see Lines 86–88).

In Fig. 1c, why does the B. subtilis extract degrade 3'3'-cAAA and 3'3'-cAAG, and the phage infected extracts do not?

We have corrected the legend of Figure 1 to clarify that grey boxes indicate nucleotides that were degraded by the uninfected control lysate and therefore not tested against phage-infected lysates.

*I would call “Apyc1” “Apy1” to be consistent with typical bacterial gene/protein naming (3 letters and a number) and to be consistent with anti-CRISPR (*Acr*) and anti-CBASS (*Acb*).*

We thank the reviewer for their suggestion. Pycsar immunity proteins are named beginning with “Pyc” and use 4–6 characters (*PycC*, *PycTM*, *PycTIR*, etc.) (Tal and Morehouse et al Cell 2021 PMID 34644530) so we prefer to keep the name *Apyc1* to clearly designate a role in anti-Pycsar evasion. We also note that the names for anti-CRISPR proteins are often >6 characters including the cA4 nuclease *AcrIII-1* (Athukoralage et al Nature 2020 PMID 31942067; Jia and Patel, Nat Rev Mol Cell Biol 2021 PMID 34089013).

Referee #2

This manuscript describes the discovery of virus encoded inhibitors for the recently discovered CBASS and Pycsar phage defense systems. Although virus encoded nucleases for signalling molecules have been found before against other virus defense systems (CRISPR, cGAS-STING), this is an interesting and important manuscript that helps to further understand CBASS and Pycsar, and its escape by phage. Although the paper is clearly written, the assembly and order of results is somewhat unusual because of two systems investigated here, and the broad combination of different techniques used, making the paper somewhat unfocused. Still the data is of high quality and the insights obtained are broad. Two points need to be addressed.

We are glad the reviewer found our data to be of high quality and broad importance to the field, and we thank them for their helpful comments to further improve our manuscript.

Major points

- It is unclear if the phages that naturally carry the inhibitor genes can overcome CBASS and Pycsar immunity. Do they escape immunity and to what extent?

Many phages have been shown to efficiently replicate in the presence of functional CBASS and Pycsar immune systems (Cohen et al Nature 2019 PMID 31533127; Tal and Morehouse et al Cell 2021 PMID 34644530). It is not clear if these phages avoid detection, and therefore do not activate cyclic nucleotide-mediated defenses, or if specific inhibitors are responsible for enabling immune evasion. To address this question and extend our discovery of the first anti-CBASS and anti-Pycsar proteins, we present new virology experiments demonstrating that replication of phage T4 in the presence of active CBASS defense is specifically dependent on the activity of *Acb1* (see response to next point below).

*- Further, in order to demonstrate that phage use *Acb1* and *Apyc1* genes to evade CBASS and Pycsar defense systems it would be necessary to knock-out (or knock-down) the inhibitor genes in the natural phage. Now only the introduction of *Acb1* and *Apyc1* is shown in trans and this is tested with phage P1 and T5 that naturally do not contain these inhibitors. The suggested experiment will determine if these inhibitors actually contribute to a phenotype.*

We thank the reviewer for this suggestion. To address the reviewer’s point, we focused on the well-characterized phage T4 and performed extensive new experiments to define the *in vivo* importance of cyclic nucleotide-specific nucleases in immune evasion. First, we adapted a recently developed CRISPR/Cas9-based strategy (Tao et al, PMID 28657724) to introduce nonsense mutations and engineer a mutant phage T4 Δ *Acb1* virus (Extended Data Figure 10a). *E. coli* cells infected with phage T4 Δ *Acb1* exhibit no ability to hydrolyze 3'3'-cGAMP or other cyclic nucleotides, confirming that *Acb1* is essential for viral degradation of CBASS immune signals (Extended Data Figure 10b). We next compared growth of wildtype phage T4 and phage T4 Δ *Acb1* in *E. coli* encoding functional or

catalytically inactive CBASS immune systems. In the absence of functional CBASS defense, phage T4 and phage T4 Δ Acb1 grow equally well and release 10^{11} particles mL^{-1} revealing that Acb1 is not required for normal replication in *E. coli*. In contrast, growth of phage T4 Δ Acb1 is specifically impaired in the presence of active CBASS immunity with the mutant virus exhibiting a $>1,000$ fold defect in viral replication compared to wildtype phage T4 (Figure 4e,f; Extended Data Figure 10c,d). Combined with our previous experiments showing Acb1 expression rescues growth of phage P1 from CBASS-mediated inhibition (Figure 4c), these results demonstrate that viral nucleases are necessary and sufficient for evasion of cyclic nucleotide-mediated phage defense.

Our results demonstrate that expression of Apyc1 is sufficient to rescue growth of phage T5 and enable evasion of Pycsar immunity (Figure 4d). As the technology necessary for *B. subtilis* phage engineering is poorly developed, we note in the text that our knock-out studies focus on phage T4 (see Lines 173–174). Extending understanding of the specific role of Apyc1 in phage SBSphiJ anti-Pycsar evasion is a goal for future experiments.

Referee #3

In their manuscript, Hobbs et al. describe the function, structure and mechanisms of phage-encoded proteins that allow phages to evade CBASS and Pycsar immunity.

First, the authors incubate phage-infected culture lysates with a large array of cyclic nucleotides that could be potential signalling molecules involved in prokaryotic immunity. This shows that various phages encode proteins that are able to degrade one or more (types of) signalling molecules. Next, they identify and two suspect enzymes, Acb1 and Apyc1, and show the phylogeny of these enzymes which are encoded in a plethora of phages. Furthermore, it is demonstrated that these enzymes can degrade different types of signalling molecules in vitro, confirming they are the enzymes responsible for degradation of the signalling molecules. Next, the authors have solved the structures of apo-Acb1, substrate-bound Acb1, and apo-Apyc1. Structure-guided mutants are made and provide insights into the catalytic mechanism and substrate specificity of these enzymes. Finally, the authors demonstrate both in vivo and in vitro that Acb1 (but not Apyc1) can prevent Cap5 activity in presence of 3'3'-cGAMP, while Apyc1 (but not Acb1) can prevent PycTIR activity in presence of cUMP, implying their activity specifically prevents activation of each immune system.

The methodology is appropriate, the data supports the novel and exciting conclusions drawn, and the manuscript extremely well written. I only have a couple of suggestions.

We thank the reviewer for highlighting our conclusions as novel and exciting, and we are grateful for their helpful feedback to improve our manuscript.

1. The authors present Apyc1 and Acb1 as anti-Pycsar and anti-CBASS enzymes in their manuscript. However, for their function, these enzymes might not have to specifically interact with Pycsar and CBASS-system proteins (as certain antiCRISPR proteins do). Have the authors investigated if there are interactions between these immune systems and the characterized proteins? Apyc1 and Acb1 degrade messenger molecules that theoretically could also be used by other (to-be-characterized) prokaryotic immune systems. For example, certain type III CRISPR systems are known to (also) generate cA3 molecules recognized by Acb1 ((Gruschow et al 2019, Kazlauskienė et al, 2017, Han et al 2018). While most characterized effectors rely on other cA# molecules for their activation, it is also known that many putative effectors exist, and they might be regulated by such signalling molecules. Therefore, it might well be that Apyc1 and Acb1 also interfere with other immune systems. The authors should at least reflect on this.

We appreciate the reviewer's point and agree that overlap between immune signals is an important contribution to the host-pathogen interactions governing immune evasion of cyclic nucleotide-mediated defense. All our data support that Acb1 and Apyc1 directly target specialized cyclic nucleotide immune signals and do not require specific interaction with CBASS or Pycsar proteins. As most of the cyclic nucleotides targeted by Acb1 and Apyc1 are unique signals not shared with type III CRISPR systems, our data suggest that the primary function of these enzymes is to enable evasion of CBASS/Pycsar defense. Additionally, we specifically tested cA4 molecules characteristic of type III CRISPR immunity and observed that Acb1 and Apyc1 are unable to hydrolyze larger cyclic nucleotide species (Extended Data Figure 7d). In our revised manuscript we now further discuss these findings and include the

reviewer's point that as some type III CRISPR systems produce cyclic trinucleotide signals it is possible that an added benefit of anti-CBASS and anti-Pycsar nuclease enzymes may be the ability to evade additional defense systems (see Lines 194–196).

2. Almost all in vitro assays and in vivo experiments lack a negative control, preferentially a sample in which a catalytic mutant of the enzyme is used. This should at least be added for the in vivo experiments (to confirm the observed effects are a consequence of Acb1/Apyc1 activity, and not just because of their expression). Otherwise, it cannot be ruled out that these proteins can inhibit the systems in another way (e.g. by direct binding).

To address the reviewer's point, we have repeated the effector inhibition assays with catalytic inactive proteins as additional negative controls. Acb1 and Apyc1 proteins with mutations to the catalytic active site residues lose all ability to prevent CBASS and Pycsar effector activation (Figure 4a–b; Extended Data Figure 9a,b,e).

As additional controls for the importance of phage nuclease enzymes *in vivo*, our revised manuscript now includes significant new virology experiments characterizing the specific role of Acb1 during phage infection. First, we adapted a recently developed CRISPR/Cas9-based strategy (Tao et al, PMID 28657724) to introduce nonsense mutations and engineer a mutant phage T4 Δ Acb1 virus (Extended Data Figure 10a). *E. coli* cells infected with phage T4 Δ Acb1 exhibit no ability to hydrolyze 3'3'-cGAMP confirming that Acb1 is essential for viral degradation of CBASS immune cyclic nucleotides (Extended Data Figure 10b). We next compared growth of wildtype phage T4 and phage T4 Δ Acb1 in *E. coli* encoding functional or catalytically inactive CBASS immune systems. In the absence of functional CBASS defense, phage T4 and phage T4 Δ Acb1 grow equally well and release 10^{11} particles mL⁻¹ revealing that Acb1 is not required for normal replication in *E. coli*. In contrast, growth of phage T4 Δ Acb1 is specifically impaired in the presence of active CBASS immunity with the mutant virus exhibiting a >1,000-fold defect in viral replication compared to wildtype phage T4 (Figure 4e,f; Extended Data Figure 10c,d). Combined with our previous experiments showing Acb1 expression rescues growth of phage P1 from CBASS-mediated inhibition (Figure 4c), these results demonstrate that viral nucleases are necessary and sufficient for evasion of cyclic nucleotide-mediated phage defense.

4. It should be discussed in the introduction that E. coli strains naturally encode Pycsar and CBASS systems. This adds to the significance that E. coli phages encode inhibitors of this system. Are the systems also found in Bacillus subtilis strains? This should be discussed, also when this is not the case.

We thank the reviewer for this point. We have clarified in the text that CBASS and Pycsar systems are encoded in *E. coli*, *B. subtilis*, and diverse bacterial species (Cohen et al Nature 2019 PMID 31533127; Tal and Morehouse et al Cell 2021 PMID 34644530) (see Lines 24–25).

5. The R values for the Apyc1 structure are (too) high for a 2.7Å structure. Furthermore, there is a high amount of Ramachandran and sidechain outliers. This structure should be refined further, or an explanation should be given for the poor statistics. Once the structure has been further refined, I would recommend to add the structure-function characterization of Apyc1 to Figure 3 too, as it is currently underrepresented.

We thank the reviewer for closely checking the statistics of our crystallography data. The Bsp38 Apyc1 structure statistics are within the acceptable range of Protein Data Bank values but we agree that the R values are higher than anticipated for a 2.7 Å structure. Two limitations to our ability to further refine the Bsp38 Apyc1 structure are comparably lower quality diffraction data for this crystal structure and that the initial phase solution was derived using molecular replacement and a modestly accurate model prepared with AlphaFold2 (Jumper et al Nature 2021 PMID 34265844). In spite of significant effort, we have been unable to collect anomalous data for experimental phase information or higher-quality diffraction data for Bsp38 Apyc1. As new experimental data to support structural understanding of Apyc1 activity, we have determined crystal structures of two closely related bacterial *Paenibacillus* Apyc1 enzymes identified in our Apyc1 phylogenetic analysis including a new co-crystal structure in complex with a cAMP analog (Extended Data Figure 8b–d). Notably, we were able to phase these structures with SeMet-derivatized crystals and have refined the 1.5 Å and 1.8 Å structures with significantly improved R value statistics (SI Table 1). The new *Paenibacillus* structures confirm close homology with phage Bsp38 Apyc1 and conservation of an open active-site architecture and an extended Apyc1-specific active-site loop that support cNMP hydrolysis independent of nucleobase

identity (Extended Data Figure 8b–f). Together, our new results confirm that both phage and *Paenibacillus* Apyc1 enzymes rapidly degrade cCMP/cUMP and exhibit activity that is unique compared to other previously characterized cNMP phosphodiesterases.

We agree with the reviewer that additional Apyc1 structural data would be important to include in the main text but unfortunately space is not available within the constraints of the journal's formatting guidelines. To better feature these data, we have revised the Extended Data to create a figure dedicated to analysis of the Bsp38 and *Paenibacillus* Apyc1 structures (Extended Data Figure 8).

6. The authors claim that Apyc1 has a broad specificity, but only show it degrades cNMPs (in contrast to Acb1, for which they show degradation of different types of cyclic nucleotides). This suggests its activity is selective for very similar cNMPs (and not broad as claimed) - it is just not selective for specific bases on cNMPs. Does Apyc1 also degrade other types of (cyclic nucleotide) substrates? Based on Figure 1 I would guess this is not the case, and I would rephrase the claim to something like 'it degrades cNMPs with no selection for specific bases'.

As new data in our revised manuscript we tested the activity of Apyc1 against alternative nucleotide substrates and observed that Apyc1 has no ability to degrade non-cNMP substrates including 3'3'-cGAMP, 3'3'-cAA, 3'3'3'-cAAA, and NAD⁺ (Extended Data Figure 5f). We have clarified the text to specify that Apyc1 functions as a cNMP hydrolyase and is able to target specialized cCMP and cUMP signals used in Pycsar immunity because the enzyme does not discriminate against nucleobase identity.

7. It is unclear what the error bars in Figure 2 and 4, and several extended data figures indicate.

We apologize for this omission and have corrected the figure legends to specify all error bars.

Referee #4

The authors of this manuscript identify and characterize a novel class of enzymes that specifically counter cyclic nucleotide based anti-phage defense systems. Starting from the observation that recently identified anti-phage signaling compounds like 3'3'-cGAMP or cCMP are highly stable in bacterial cell extracts, but rapidly degraded when challenged with extracts of phage-infected cells, they set out to biochemically define the respective hydrolytic activities. This led to the discovery of novel, phage-encoded hydrolytic enzymes which display broad recognition of different cyclic nucleotides and which confer the ability to overrule nucleotide-based phage defense in vitro and in vivo. Based on these findings and on the observation that members of these enzyme families are widespread in phylogenetically diverse phages, the authors conclude that degradation of host-derived cyclic nucleotides represents a common immune evasion strategy of phages.

This work is clearly exciting and novel and of broad relevance as it adds additional components of phage immune evasion thereby contributing to a better understanding of bacterial innate immunity. Experiments are well executed and convincing. I only have a few inquiries and suggestions.

We are glad the reviewer found our work exciting, novel, and of broad relevance, and we thank them for their helpful feedback.

It is not clear why Apyc1 was discovered specifically with phages infecting B. subtilis. The authors recently demonstrated that E. coli strains do encode Pycsar immunity components. However, none of the E. coli phages seems to be equipped with nucleases targeting Pycsar. Please comment.

We have clarified in the text that CBASS and Pycsar systems are encoded in *E. coli*, *B. subtilis*, and diverse bacterial species (Cohen et al Nature 2019 PMID 31533127; Tal and Morehouse et al Cell 2021 PMID 34644530) (see Lines 24–25). We agree that it is interesting that cCMP/cUMP cleavage activity was only detected in lysates infected with *Bacillus* phages. We have conducted a revised bioinformatic analysis of Apyc1 distribution and observe that closely related enzymes are currently found primarily in *Bacillus* phages (Figure 2j), suggesting that future study of additional *E. coli* phages may identify distinct mechanisms used to evade Pycsar defense.

Fig. 2i,j The figures display the distribution of Acb1 and Apyc1 homologs without providing specific information about the respective phylogeny. It would be valuable for the reader to have some additional information to understand how broadly these enzymes are distributed and if most (all?) of them are associated with prophages.

We thank the reviewer for this suggestion. We have revised our bioinformatic analysis and improved presentation of the data by including separate panels for analysis of viral proteins (Figure 2j) and closely related bacterial homologs (Extended Data Figure 6b). Greater than 97 percent of Acb1 homologs are encoded in phages and integrated prophage sequences, whereas Apyc1 homologs encoded in >100 phage but are also prevalent in some bacterial genomes. Whether these bacterial Apyc1 homologs are located within cryptic prophages or are involved in host regulation of cyclic nucleotide levels is an exciting opportunity for future study.

Fig. 3 The specificity of Acb1 for cyclic nucleotides containing at least one adenine moiety is intriguing. It is also interesting that specificity for A is mediated by the same residue (E141) that rotates into the active site upon substrate binding, apparently inflicting a rotation of the adenine base and a stretched substrate conformation. Is it possible that the observed limited substrate specificity of Acb1 is dictated by the catalytic mechanism of these enzymes? Or in other words, do substrates require at least one A and specific base contact because a strained substrate conformation is critical for catalysis? The observation that the E141A mutant variant has retained activity against cGAMP in vitro, argues that this is not the case, but no kinetic data are presented. It would be very interesting to investigate if the E141A mutant no longer discriminates against non-As like cGG.

As new data in our revised manuscript we compared the ability of wildtype Acb1 and Acb1 E141A to degrade 3'3'-cGAMP, 3'3'-cAA, and 3'3'-cGG. These results demonstrate that the E141A mutation causes a slight decrease in ability to cleave 3'3'-cAA and a slight increase in the ability to cleave 3'3'-cGG compared to the WT enzyme (Extended Data Figure 7c). Based on these data, it appears that E141 is only partially responsible for the specificity of Acb1 for adenine-containing nucleotides. Base-stacking interactions between adenine and Y12 and W174 of Acb1 play a more significant role in catalysis (Figure 3e). Thus, it is likely the combination of these stacking and electrostatic interactions that leads to the specificity of Acb1.

In the last paragraph of the manuscript the authors speculate that diversification of cyclic nucleotide signals is a key host adaptation to maintain anti-phage defense (l.166) and that viral nucleases have a prominent role in driving the evolution of cyclic nucleotide mediated immune defense systems (l.181). This appears to make little sense, given the very broad substrate specificities of phage-encoded hydrolytic activities. Please specify.

We thank the reviewer for this point. We have clarified the text to state that our data support that dedicated enzymes are required to hydrolyze specific classes of cyclic nucleotide signals. As there is no single solution for degrading cNMPs (Pycsar), cyclic di/tri-nucleotides (CBASS), and larger cA4–cA6 cyclic oligonucleotides (type III CRISPR), evolving defense systems that use distinct classes of cyclic nucleotide signals creates an opportunity for bacteria to participate and successfully compete in an arms race with phages. Rare CBASS nucleotides including 3'3'-cUU and 3'2'-cGAMP resist Acb1 degradation (Figure 1a; Extended Data Figures 1 and 2), suggesting that it is possible for host defense systems to use specific nucleotide signals that can overcome phage evasion mechanisms.

l.177 'The astounding diversity...'. This statement is unclear. Please clarify.

We have corrected the sentence to read "The large diversity of >180 possible nucleotide signals proposed to exist in anti-phage defense suggests that in addition to signal degradation phages may encode Acb and Apyc proteins that target alternative components of CBASS or Pycsar immunity." (see Lines 199–202).

Reviewer Reports on the First Revision:

Referees' comments:

Referee #1 (Remarks to the Author):

In this revised manuscript, the authors have performed a large number of new experiments and have effectively addresses the key reservations that I had with the initial submission. Most important, the authors mutated the *acb1* gene in phage T4 and showed that this mutant phage could no longer replicate on a bacterial strain expressing an active CBASS system. This experiment proves that the *acb1* gene functions as an anti-immune system protein within the natural context of a phage genome. With this additional experiment and the others added, this manuscript is now certainly acceptable for publication in Nature.

I have only one reservation left that the authors could address with a small amount of re-wording. I expect that the proteins of the *Apyc1* family are also phage anti-immune system proteins, but the authors have not performed the rigorous proof that was performed with the T4 *acb1* gene. This is fine, but I think the authors should make it clear that the rigorous proof still needs to be performed for the *Apyc1* family. So I'm asking the authors to just add this caveat somewhere. I also note that *Apyc1* homologues are not nearly as widely distributed as *Acb1* homologues. This fact alone does not argue against *Apyc1* being a bona fide phage anti-immune protein. However, the authors did not answer the question of whether phages or prophages with *apyc1* genes are likely infecting strains that possess *Pycsar* systems. It would be valuable to mention this somewhere in the paper. Also, I still wonder why there are so many *apyc1* genes in bacterial genomes that appear not to be in prophages. Do these genes appear to be in other types of mobile elements? Do they co-occur with *Pycsar* systems? Maybe *Apyc1* homologues do other things besides blocking *Pycsar*. Whatever the answers to these questions are, it would not affect my opinion that this is definitely a Nature paper. Addressing these questions would allow the reader to speculate for themselves.

Referee #2 (Remarks to the Author):

The authors have sufficiently addressed my concerns.

Referee #3 (Remarks to the Author):

The authors have addressed almost all my comments in a satisfactory manner. I recommend publication, and have two final minor textual suggestions that might aid readers:

-The authors state "We have clarified in the text that CBASS and *Pycsar* systems are encoded in *E. coli*, *B. subtilis*, and diverse bacterial species", but this is not clear to me from the text, especially not in the introduction (I only find it is encoded in diverse bacterial species, and find that *E. coli* CBASS is

used in experiments in Figure 4). It would be helpful for the reader to state specifically they are (also) found in *E. coli* and *B. subtilis*.

-The authors convincingly show that 3'3'-cGAMP is converted to GpAp by Acb1, but it is not mentioned in the manuscript text - might be nice to add.

Referee #4 (Remarks to the Author):

The authors have responded adequately to all my queries.

Author Rebuttals to First Revision:

Referee #1:

*In this revised manuscript, the authors have performed a large number of new experiments and have effectively addresses the key reservations that I had with the initial submission. Most important, the authors mutated the *acb1* gene in phage T4 and showed that this mutant phage could no longer replicate on a bacterial strain expressing an active CBASS system. This experiment proves that the *acb1* gene functions as an anti-immune system protein within the natural context of a phage genome. With this additional experiment and the others added, this manuscript is now certainly acceptable for publication in Nature.*

We appreciate the reviewer for highlighting the large number of new experiments in our revised manuscript and we thank them again for their helpful comments to improve our manuscript.

*I have only one reservation left that the authors could address with a small amount of re-wording. I expect that the proteins of the *Apyc1* family are also phage anti-immune system proteins, but the authors have not performed the rigorous proof that was performed with the T4 *acb1* gene. This is fine, but I think the authors should make it clear that the rigorous proof still needs to be performed for the *Apyc1* family. So I'm asking the authors to just add this caveat somewhere.*

Our data demonstrate that expression of *Apyc1* is sufficient to enable phage to evade *Pyocsar* defense (Figure 4), but we agree with the reviewer that genetic deletion of *Apyc1* from *B. subtilis* phages is an important direction for future research. We now include this point as a specific statement in the main text (See Lines 169–171).

*I also note that *Apyc1* homologues are not nearly as widely distributed as *Acb1* homologues. This fact alone does not argue against *Apyc1* being a bona fide phage anti-immune protein. However, the authors did not answer the question of whether phages or prophages with *apyc1* genes are likely infecting strains that possess *Pyocsar* systems. It would be valuable to mention this somewhere in the paper. Also, I still wonder why there are so many *apyc1* genes in bacterial genomes that appear not to be in prophages. Do these genes appear to be in other types of mobile elements? Do they co-occur with *Pyocsar* systems? Maybe *Apyc1* homologues do other things besides blocking *Pyocsar*. Whatever the answers to these questions are, it would not affect my opinion that this is definitely a Nature paper. Addressing these questions would allow the reader to speculate for themselves.*

We thank the reviewer for raising this additional point. Our analysis demonstrates that *Apyc1* is widely conserved in diverse phages (Figure 2) and is additionally present in bacteria (Extended Data Figure 6b). We suspect that many instances of *Apyc1* conservation in bacteria may be due to cryptic prophages present in bacterial genomes, but we agree that regulation of *Pyocsar* defense or modulation of other cNMP-signaling pathways are also intriguing possibilities for future investigation. We now highlight these possibilities in the main text (See Lines 81–84).

Referee #2:

The authors have sufficiently addressed my concerns.

We thank the reviewer again for their helpful comments to improve our manuscript.

Referee #3:

The authors have addressed almost all my comments in a satisfactory manner. I recommend publication, and have two final minor textual suggestions that might aid readers:

We thank the reviewer again for their helpful comments to improve our manuscript.

*-The authors state "We have clarified in the text that CBASS and *Pyocsar* systems are encoded in *E. coli*, *B. subtilis*, and diverse bacterial species", but this is not clear to me from the text, especially not in*

the introduction (I only find it is encoded in diverse bacterial species, and find that E. coli CBASS is used in experiments in Figure 4). It would be helpful for the reader to state specifically they are (also) found in E. coli and B. subtilis.

We apologize for confusion created by our edit in the revised manuscript. We now specifically state that CBASS and Pycsar are encoded in both *E. coli* and *B. subtilis* in the main text (See Lines 20–23).

-The authors convincingly show that 3'3'-cGAMP is converted to GpAp by Acb1, but it is not mentioned in the manuscript text - might be nice to add.

We agree with the reviewer and have included this point to the main text (See Lines 106–112).

Referee #4:

The authors have responded adequately to all my queries.

We thank the reviewer again for their helpful comments to improve our manuscript.